EMBO
Molecular Medicine

# Targeting the SHP2 phosphatase promotes vascular damage and inhibition of tumor growth

Yuyi Wang[1], Ombretta Salvucci[1], Hidetaka Ohnuki[1], Andy D Tran[2], Taekyu Ha[1], Jing-Xin Feng[1] (iD),
Michael DiPrima[1], Hyeongil Kwak[1] (iD), Dunrui Wang[1] (iD), Yanlin Yu[3], Michael Kruhlak[2] &
Giovanna Tosato[1,*] (iD)

## Abstract

The tyrosine phosphatase SHP2 is oncogenic in cancers driven by receptor-tyrosine-kinases, and SHP2 inhibition reduces tumor growth. Here, we report that SHP2 is an essential promoter of endothelial cell survival and growth in the remodeling tumor vasculature. Using genetic and chemical approaches to inhibit SHP2 activity in endothelial cells, we show that SHP2 inhibits pro-apoptotic STAT3 and stimulates proliferative ERK1/2 signaling. Systemic SHP2 inhibition in mice bearing tumor types selected for SHP2-independent tumor cell growth promotes degeneration of the tumor vasculature and blood extravasation; reduces tumor vascularity and blood perfusion; and increases tumor necrosis. Reduction of tumor growth ensues, independent of SHP2 targeting in the tumor cells, blocking immune checkpoints, or recruiting macrophages. We also show that inhibiting the Angiopoietin/TIE2/AKT cascade magnifies the vascular and anti-tumor effects of SHP2 inhibition by blocking tumor endothelial AKT signaling, not a target of SHP2. Since the SHP2 and Ang2/TIE2 pathways are active in vascular endothelial cells of human melanoma and colon carcinoma, SHP2 inhibitors alone or with Ang2/TIE2 inhibitors hold promise to effectively target the tumor endothelium.

**Keywords** cancer; endothelial cells; SHP2; tumor growth; tumor vasculature

**Subject Categories** Cancer; Vascular Biology & Angiogenesis

## Introduction

Endothelial cells, which line the internal surface of blood vessels, are structural components of the vasculature that supplies blood to tissues and are poised to respond to environmental insults. Considerable work has focused on targeting the structural functions of endothelial cells, particularly to limit the supply of blood and nutrients to cancer tissues. Neutralization of vascular endothelial growth factor (VEGF) reduces tumor neovascularization and drugs that neutralize VEGF activity have increased the survival for patients suffering from certain cancers (Apte *et al*, 2019). However, resistance to anti-VEGF treatment is common and its emergence has been attributed to the presence of endothelial cell populations which are VEGF receptor-deficient, display ligand-independent VEGF receptor activity, rely on growth factors other than VEGF and to other causes (Sitohy *et al*, 2011; Eichten *et al*, 2016; Apte *et al*, 2019). One such factor is Angiopoietin-2 (Ang2), and the combined neutralization of VEGF and Ang-2 has shown promising results in initial trials (Schmittnaegel *et al*, 2017; Apte *et al*, 2019).

The structurally and functionally disordered tumor vasculature undergoes sustained remodeling in response to changing signals from cancer cells and the tumor microenvironment (Potente *et al*, 2011). In mice, anti-VEGF responsive "sprouting" tumor vessels evolve into anti-VEGF unresponsive arterial-venous "vascular malformations" (Nagy & Dvorak, 2012). However, little is known about the mechanisms that orchestrate remodeling of post-angiogenic tumor vessels and how remodeling relates to anti-angiogenic therapy resistance.

EphrinB2 is a transmembrane ligand for Eph receptors that plays pivotal roles in vascular biology during development and after birth (Pasquale, 2008; Kania & Klein, 2016). Once tyrosine phosphorylated by either EphB receptor engagement or trans-phosphorylated by the tyrosine kinase (TK) receptors TIE2, FGFR, and PDGFR, EphrinB2 regulates cell-to-cell adhesion and cell movement (Adams *et al*, 1999; Kania & Klein, 2016). We have identified a pro-survival role for tyrosine-phosphorylated EphrinB2, which relies upon the associated phosphatase SHP2 (SRC homology 2-containing Protein Tyrosine Phosphatases (PTP)) preventing activation of a STAT-dependent pro-apoptotic pathway in endothelial cells (Salvucci *et al*, 2015). Inactivation of this pro-survival pathway is critical for the physiologic involution of hyaloid vessels in the developing eye (Salvucci *et al*, 2015).

We now hypothesized that EphrinB2/SHP2-dependent signaling plays a role in the regulation of tumor vessel survival. Here, we identify active SHP2 as an essential guardian of endothelial cells and tumor vessel persistence by simultaneously repressing STAT3

1   Laboratory of Cellular Oncology, Center for Cancer Research, National Cancer Institute, National Institutes of Health, Bethesda, MD, USA
2   Center for Cancer Research Microscopy Core, Laboratory of Cancer Biology and Genetics, National Cancer Institute, National Institutes of Health, Bethesda, MD, USA
3   Laboratory of Cancer Biology and Genetics, Center for Cancer Research, National Cancer Institute, National Institutes of Health, Bethesda, MD, USA
    *Corresponding author. Tel: +1 240 760 6144; E-mail: tosatog@mail.nih.gov

signaling and activating ERK1/2 signaling in endothelial cells. By rigorously selecting or genetically engineering tumor types composed of SHP2 growth-independent tumor cells, we find that specific SHP2 inactivation promotes endothelial cell death and the involution of tumor vessels resulting in reduced tumor growth, without impacting the resting vasculature of normal tissues. The current identification of SHP2 as a previously unrecognized regulator of the tumor vasculature delineates a novel approach to reducing tumor vascularity and tumor growth.

# Results

### Identification of a mouse model for EphrinB2/SHP2 targeting in the tumor vasculature

Based on our previous experiments showing that tyrosine-phosphorylated (p)-EphrinB2/SHP2 signaling delays the physiological involution of hyaloid vessels by blocking pro-apoptotic STAT signaling, we now examined whether EphrinB is active in the tumor vasculature and contributes to preserving the viability of tumor vessels. To this end, we first searched for a tumor model in the immunocompetent mouse in which the tumor vasculature is broadly p-EphrinB-positive, but the tumor cells and other tumor-infiltrating cells are p-EphrinB-negative. Such model would help dissect the role of active EphrinB2 in the tumor vasculature.

Among the six mouse models analyzed, we found that p-EphrinB is present in most (89%) $CD31^+$ endothelial cells of B16F10 melanoma, whereas most (98%) tumor cells, pericytes, and other cells in the tumor microenvironment are p-EphrinB-negative (Fig 1A–D). The other five mouse cancer models differed from the B16F10 model in having a lower proportion (range 12–85%) of tumor endothelial cells displaying p-EphrinB (Lewis lung carcinoma/LLC1, 50%; 4T1, 25%; 9013BL/M2, 85%; Mel114433/M1 melanoma, 68% (Marie *et al*, 2020), and MOPC-315 plasmacytoma, 32%). The mammary 4T1 tumors additionally displayed p-EphrinB in most (86%) infiltrating $Gr1^+$ myeloid cells (Appendix Fig S1A and B; Salvucci *et al*, 2009). Thus, among the six models analyzed, the B16F10 mouse tumor model fulfills the criteria of displaying broadly active EphrinB in the vascular endothelial cells but not in the tumor cells and other infiltrating cells.

EphrinB2 is normally activated by receptor engagement, particularly by EphB4 (Kania & Klein, 2016). Previously, we determined that EphB4 protein is present in B16F10 tumor cells (Kwak *et al*, 2016). To test if B16F10-associated EphB4 activates tumor vascular EphrinB2, we generated syngeneic B16F10 tumors depleted of EphB4 and control tumors by infecting B16F10 cells with either a validated shRNA that effectively depletes EphB4 (Salvucci *et al*, 2015) or with the control vector (pLKO) (Fig 1E and F). The vasculature within these EphB4-negative tumors was p-EphrinB$^+$ similar to the pLKO controls (Fig 1G). In addition, robust expression of the EphB4/EphrinB2 blocking TNYL-RAW peptide (Kwak *et al*, 2016) did not reduce vascular p-EphrinB in B16F10 tumor-bearing mice (Appendix Fig S1C and D). Among the other EphB receptors that can activate EphrinB2, B16F10 expresses low levels of EphB6 and EphA4. These experiments indicated that EphBs in B16F10 cells are unlikely responsible for EphrinB activation in the B16F10 tumor vasculature.

In addition to becoming phosphorylated through binding interaction with EphB receptors and subsequent Src kinases recruitment, the EphrinB1 intracellular domain is trans-phosphorylated in cis by certain TK growth factors receptors, including TIE2, FGFR, and PDGFR (Bruckner *et al*, 1997; Adams *et al*, 1999; Kania & Klein, 2016). We broadly examined the potential of TK receptors to trans-phosphorylate EphrinB2 in endothelial cells. Using a network algorithm (GPS 3.0), we identified 48 unique receptor kinases/kinases with the potential to phosphorylate EphrinB2 (residues 251–322), only 21 of which are significantly expressed in murine dermal microvascular endothelial cells (Fig EV1A). To prioritize among these 21 candidates, we examined differential gene expression among the three murine cancer cell lines (B16F10, LLC1, and 4T1) only one of which (B16F10) generates tumors with widespread EphrinB2 phosphorylation in the vasculature. Among the 481 differentially expressed genes, 32 code for soluble factors/transmembrane proteins, the likely activators of receptor kinases in endothelial cells. Among these, 12 were identified as directly or indirectly capable of phosphorylating EphrinB2; among them Angiopoietin-2 (Ang2) the ligand of TIE2 (Fig EV1B). We found that B16F10 cells express substantially higher mRNA levels of the TIE2 ligand Angiopoietin-2 (Ang2) compared to LLC1 and 4T1 tumor cells, whereas Ang1, Platelet-Derived Growth Factor Alpha (PDGFA), and Fibroblast Growth Factor2 (FGF2) are similarly expressed (Fig EV1C). B16F10 cells also secrete Ang2 in the culture supernatant (Fig EV1D) and B16F10 tumor-bearing mice have significantly more circulating Ang2 than control mice (Fig EV1E). In addition, Ang2 is detected in B16F10 tumor tissues, particularly in the cytoplasm of the tumor cells (Fig EV1F).

We therefore tested whether Ang2 can activate EphrinB2 in endothelial cells through its TIE2 TK receptor (Thurston & Daly, 2012). Recombinant Ang2 dose and time-dependently induced $TIE2^{Y992}$, $EphrinB2^{Y324/329}$, and $AKT^{Ser473}$ phosphorylation in primary human umbilical vein endothelial cells (HUVEC, Fig 1H). Ang2 activated TIE2 with delayed kinetics compared to other TK receptors in HUVEC, such as VEGFR2 activation by VEGF-A (Fig EV1G); however, a similarly delayed Ang2-TIE2 activation was previously observed in HUVEC (Bogdanovic *et al*, 2006). These results suggested that p-TIE2 may contribute to activation of EphrinB2 in the tumor vasculature of B16F10 tumors. Consistent with this possibility, we found that TIE2 is broadly tyrosine phosphorylated in the $CD31^+$ endothelial cells (94.5%) of B16F10 tumors (Fig 1I). Additionally, we found that the $CD31^+$ endothelial cells of B16F10 tumors are generally p-SHP2$^{Tyr542}$ positive (78.5%), whereas the tumor cells are largely (96%) p-TIE2 and p-SHP2 negative (Fig 1I). By comparison, p-SHP2$^{Tyr542}$-positive endothelial cells represent a variable proportion of endothelial cells within LLC1, 4T1, 9013BL, and Mel114433 tumors (7, 11, 80, and 85%, respectively; Appendix Fig S1A). These results indicated that the B16F10 mouse tumor model provides an opportunity for selectively targeting p-EphrinB/SHP2 in the tumor vasculature and to investigate relationships between p-EphrinB/SHP2 and Ang2/TIE-2 signaling.

### EphrinB2/SHP2 and Ang2/TIE2 activity in vascular endothelial cells of human cancers

A screen of human melanoma and colon carcinoma biopsy specimens showed that most $CD31^+$ endothelial cells are p-EphrinB$^+$ (86

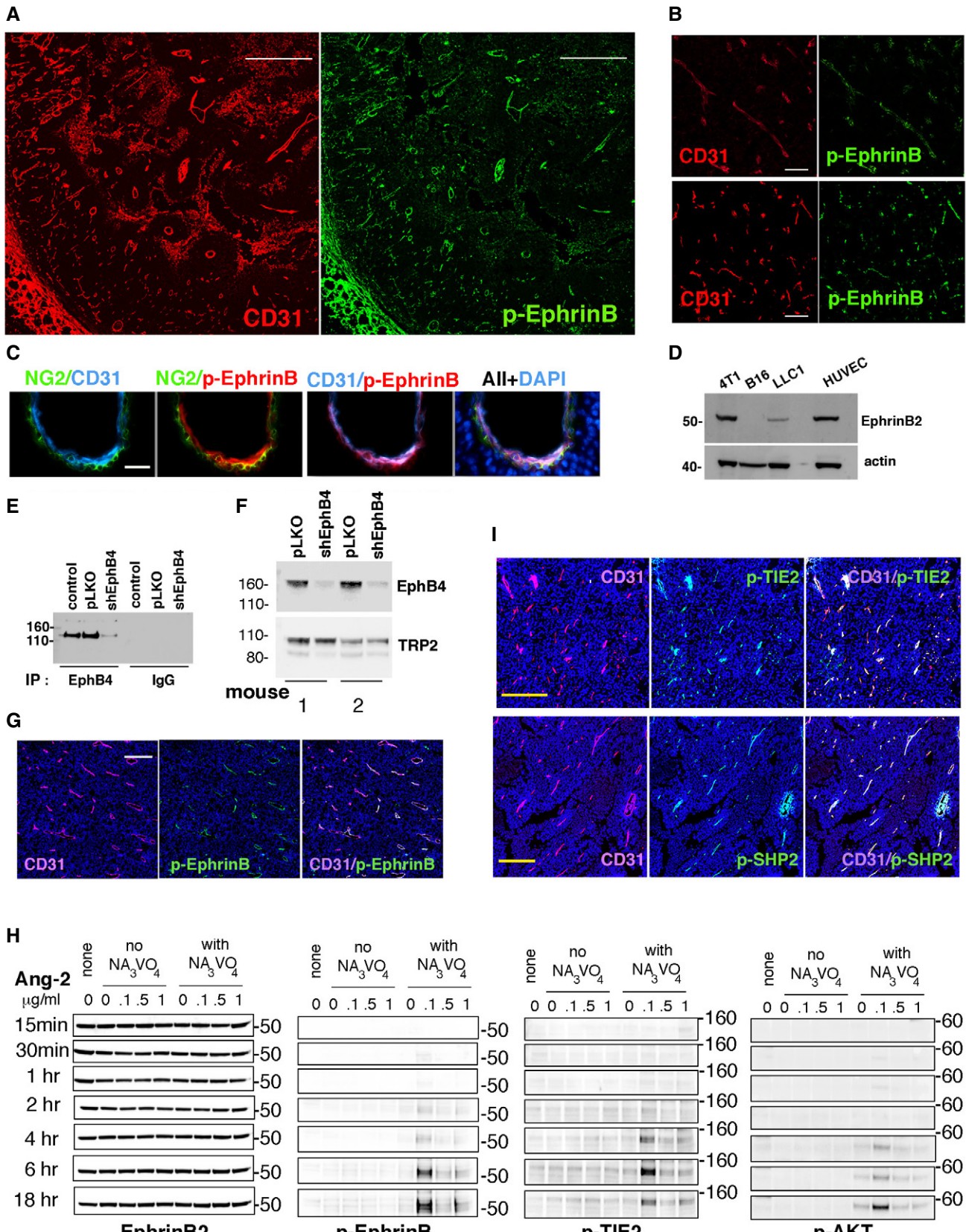

**Figure 1.**

**Figure 1.   EphrinB2, TIE2, and SHP2 phosphorylation in mouse tumor vessels.**

A, B   Representative B16F10 tumor established in a syngeneic mouse showing that tumor vessels, identified by CD31 immunostaining, are generally p-EphrinB$^+$ (scale bars: 500 μm in A; 50 μm in B).

C   CD31$^+$ endothelial cells lining a vessel are mostly p-EphrinB$^+$ whereas NG2$^+$ perivascular cells are mostly p-EphrinB$^-$; cell nuclei are identified by DAPI; confocal images (scale bar: 10 μm).

D   EphrinB2 is detected in cell lysates of 4T1, LLC1, and HUVEC, but not in B16F10 cells; representative of 3 immunoblotting.

E   EphB4 depletion in B16F10 cells; immunoprecipitation/immunoblotting; control: no infection; pLKO: infection with control lentivirus; shEphB4: infection with silencing lentivirus.

F   B16F10 tumors induced by inoculation of EphB4-depleted B16F10 cells contain reduced levels of EphB4 compared to control pLKO-infected B16F10; representative immunoblotting (of 16 mice); TRP2: tyrosine-related protein 2 (TRP2).

G   CD31$^+$ vessels in EphB4-depleted tumors are generally p-EphrinB$^+$; scale bar: 200 μm.

H   Ang-2 dose and time-dependently activates p-EphrinB, p-TIE2, and p-AKT in HUVEC; immunoblotting results; NA$_3$VO$_4$: sodium orthovanadate.

I   Tumor vessels in B16F10 tumors are mostly p-TIE2 (Tyr$^{992}$)$^+$ and p-SHP2(Tyr$^{542}$)$^+$; confocal images (scale bars: 200 μm).

Source data are available online for this figure.

and 94%, respectively), p-SHP2$^+$ (68 and 76%, respectively), p-TIE2$^+$ (83 and 82%, respectively), and Ang2$^+$ (92 and 83%, respectively) (Fig 2A). Instead, in human colon adenomas, the CD31$^+$ endothelial cells are mostly negative for p-EphrinB (84%), p-SHP2 (83%), and p-TIE2 (81%); a proportion of the CD31$^+$ endothelial cells is Ang2$^+$ (48%), (Fig 2A). Specimens from human breast adenocarcinoma, lung carcinoma, glioblastoma, angiosarcoma, and Kaposi's sarcoma showed the presence of p-EphrinB (range 12–54%), p-SHP2 (14–58%), and p-TIE2 (16–61%) in a proportion of tumor vascular endothelial cells and Ang2 in a proportion of the CD31$^-$ tumor cells (2–63%) (Abdul Pari et al, 2020).

These results indicated that the EphrinB2/SHP2 and Ang2/TIE2 pathways may be active in the tumor vasculature of human melanoma and colon carcinoma. We therefore examined whether the degree of EphrinB2 and TIE2 activity in the tumor vasculature of melanoma and colon carcinoma impacts patient probability of survival. Since Ang2 drives TIE2 and EphrinB activation in endothelial cells, we focused on Ang2 expression levels in human melanoma and colon carcinoma tissues. In addition, since active PDGFR and FGFR have the potential to trans-phosphorylate EphrinB2, we also examined expression of their specific ligands PDGFA and FGF2.

*ANGPT2* gene expression is significantly higher ($P < 0.01$) in skin melanoma compared to normal skin, whereas expression of *ANGPT1*, *PDGFA*, *FGF2*, and *VEGF-A* is not significantly different (Fig 2B). In addition, high *ANGPT2* or high *VEGF-A* expression in primary melanoma predicts a lower probability of survival compared to low *ANGPT2* ($P = 0.0004$) or *VEGF-A* ($P = 0.0052$) expression. Together, high *ANGPT2* plus high *VEGF-A* expression also predicts a significantly reduced probability ($P = 0.0002$) of survival compared to low Ang2 plus low *VEGF-A* expression (Fig 2C). However, high or low expression of *ANGPT1*, *PDGFA*, and *FGF2* has no impact on the survival probability of patients with primary melanoma.

In colon carcinoma, *ANGPT2* and *VEGF-A* expression is significantly higher ($P < 0.0001$) in the tumor compared to the adjacent normal colon, whereas expression of *ANGP1*, *PDGFA*, and *FGF2* is significantly lower ($P < 0.0001$) (Fig 2B). In patients with colon carcinoma ($n = 262$), high *ANGP2* and *VEGF-A* expression, individually, predicts a somewhat worse probability of survival compared to low *ANGPT2* or *VEGF-A* expression, whereas high or low *ANGPT1*, *PDGFA*, and *FGF2* expression has no impact (Fig 2D). Together, high *ANGPT2* plus high *VEGF-A* expression predicts a significantly ($P = 0.02$) lower probability of survival in patients with colon carcinoma (Fig 2D).

These results indicate that the EphrinB/SHP2 and Ang2/TIE2 pathways are active in the vasculature of human melanoma and colon carcinoma and that high expression of *ANGPT2* alone or with high *VEGF-A* in these cancers predicts a worse patient survival probability.

## Inhibition of SHP2 tyrosine phosphatase compromises endothelial cell viability

To evaluate the impact of SHP2-regulated signaling on endothelial cell survival, we utilized the allosteric SHP2 inhibitor SHP099, which stabilizes SHP2 in an auto-inhibited conformation and renders it enzymatically inactive (Chen et al, 2016; Garcia Fortanet et al, 2016). SHP099 dose-dependently reduces the proliferation of HUVEC, mouse bone marrow endothelial cells (BMEC) and human dermal microvascular endothelial cells (HDMEC); it also inhibits proliferation of the murine 4T1 and LLC1 cancer lines (Fig 2E). However, SHP099 (1–20 μM) minimally affects the proliferation of murine melanoma B16F10 cells (Fig 2E).

Further, we tested the effects of SHP2 silencing on endothelial and B16F10 cell proliferation. In validation experiments, 6 SHP2 shRNAs effectively depleted SHP2 from BMEC and B16F10 cells (Fig 2F). We found that SHP2 silencing effectively inhibited endothelial cell (BMEC) proliferation (Fig 2G). This effect is consistent with previous observation showing that SHP2 depletion from primary endothelial cells reduces cell proliferation, induce cell death (Mannell et al, 2008), and destabilize endothelial cell junctions reducing endothelial cell barrier functions (Zhang et al, 2019). However, SHP2 depletion from B16F10 cells did not reduce B16F10 cell proliferation (Fig 2G). This growth resistance was sustained after long-term propagation of a SHP2-depleted B16F10 cell line.

Additional experiments showed that SHP099 inhibits endothelial cell proliferation after 24–48 h incubation (Fig EV2A) with no or minimal change in cell cycle distribution (Figs 2H and EV2B). After 3-day incubation, SHP099 induces cell death in HUVEC (5 μM), HDMEC (5 μM), and BMEC (20 μM) (Figs 2I and EV2C); residual endothelial cells resume growth after wash-removal of SHP099 with kinetics comparable to control cells (Fig EV2D). However, SHP099 does not reduce the viability of B16F10 cells even when the cells are cultured in 2–20% Matrigel, under conditions that have revealed increased sensitivity to SHP099 in certain KRAS-mutant cancer cells (Hao et al, 2019) (Fig EV2E).

In endothelial cells, SHP099 induces the accumulation of cell-death-associated proteins, including cleaved caspase-3, and the

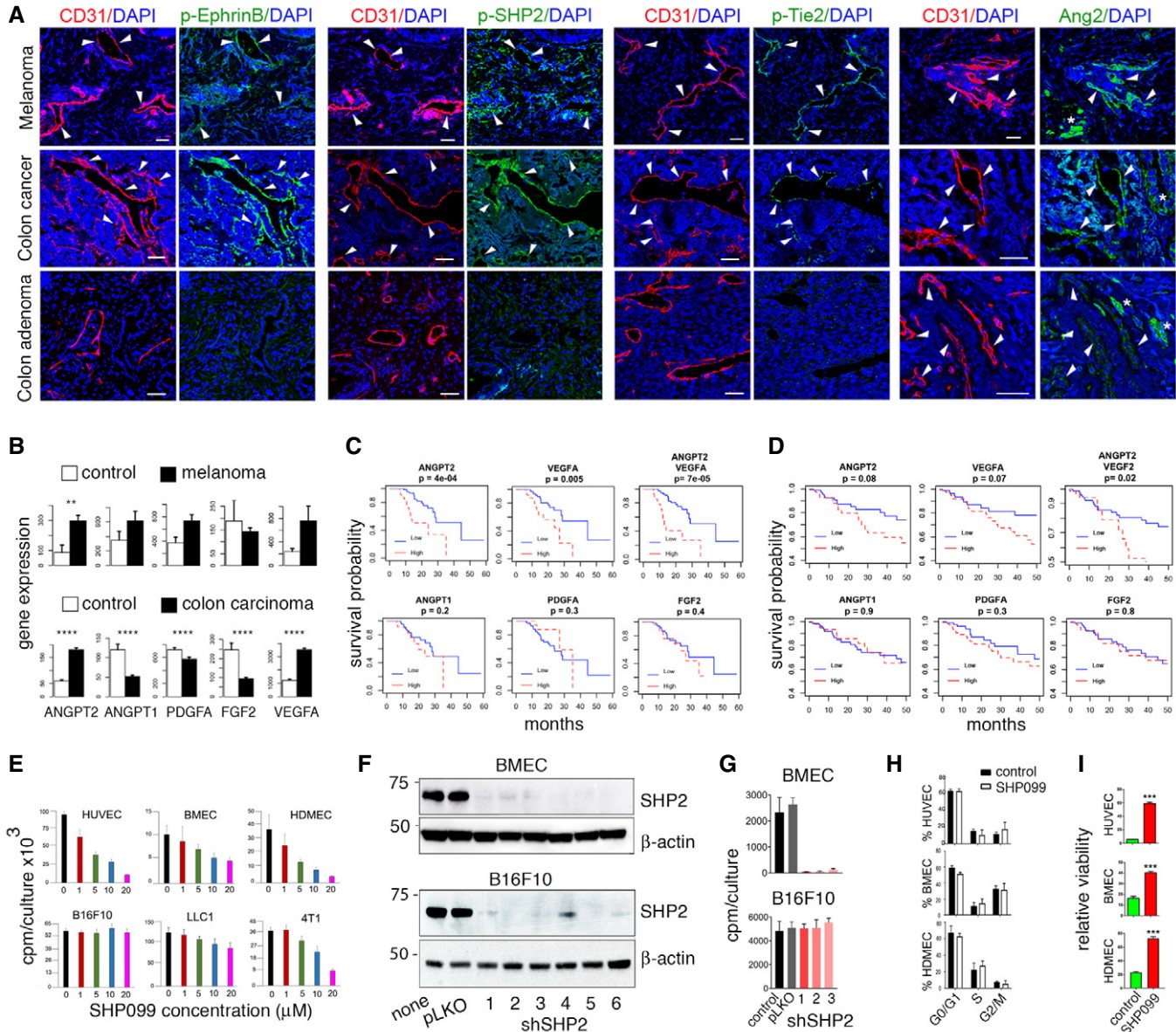

**Figure 2. EphrinB2, TIE2, and SHP2 phosphorylation in human melanoma and colon carcinoma. SHP2 inhibition in endothelial cells.**

A Representative tissue immunostaining of human melanoma, colon carcinoma, and colon adenoma; confocal images from serial sections (scale bars: 50-μm tissues). Arrowheads point to $CD31^+/p$-$EphrinB^+$; $CD31^+/p$-$SHP2^+$; $CD31^+/p$-$Tie2^+$, and $CD31^+/Ang2^+$ endothelial cells.

B Gene expression in tumor (black bar) and control (white bar); top panel: melanoma (GSE7553, primary and metastatic; tumor $n = 54$; control skin $n = 5$); bottom panel: colon cancer (TCGA, primary and metastatic; tumor $n = 269$; control $n = 41$. $**P < 0.01$, $****P < 0.0001$; $P$ values from Mann–Whitney $U$-test.

C Impact of high and low (75% quartile) mRNA levels on the survival probability of patients with primary melanoma; Kaplan–Meier analysis using TCGA primary melanoma datasets ($n = 103$); $P$ values from Mann–Whitney $U$-test.

D Impact of high and low (median) mRNA levels on the survival probability of patients with colon cancer; Kaplan–Meier analysis using TCGA colon cancer dataset ($n = 262$).

E Proliferation of endothelial (HUVEC, BMEC, and HDMED) and tumor (B16F10, LLC1, and 4T1) cells in 3-day cultures supplemented with SHP099. Representative experiments of 3–8; error bars: $\pm$ SD; 5–10 replicate cultures/experiment.

F SHP2 depletion in BMEC and B16F10 by 6 distinct shRNAs (#1-#6); pLKO: vector; none: no infection. Immunoblotting with SHP2 antibody followed by reblotting for β-actin.

G SHP2 shRNAs #1, #2, #3 reduce BMEC, not B16F10 cell proliferation; representative of three experiments; error bars: $\pm$ SD; five replicate cultures/experiment.

H SHP099 (5 μM HUVEC and HDMEC; 20 μM BMEC) minimally alters endothelial cell cycle distribution measured by flow cytometry; 3-day culture; 3 independent experiments; error bars: $\pm$ SD.

I Non-viable endothelial cells identified by flow cytometry ($DRAQ5^+/DAPI^+$) after 3-day culture with or without SHP099 (5 μM HUVEC and HDMEC; 20 μM BMEC); quantification of three independent experiments; error bars: $\pm$ SD. $P$ values from two-tailed Student's $t$-test; $***P < 0.001$.

Source data are available online for this figure.

reduction of pro-survival proteins, including Survivin (Fig EV2F). Also, SHP099 reduces endothelial cell content of vascular endothelial (VE)-cadherin (Fig EV2G), an endothelial pro-survival protein (Carmeliet *et al*, 1999) identified as a mediator of SHP2 protective function in endothelial cells (Zhang *et al*, 2019). Furthermore, SHP2 increases the levels of the pro-apoptotic JNK3 in BMEC and HUVEC (Fig EV2H). These results show that SHP099 reduces endothelial cell proliferation and induces cell death in endothelial cells.

## SHP2 modulates STAT and MAPK signaling in endothelial cells

The SHP2 phosphatase has many substrates and has been linked to regulation of several signaling cascades, including the ERK/MAPK and STAT cascades (Chan *et al*, 2008; Barr *et al*, 2009). To identify the mechanisms underlying SHP099 inhibition of endothelial cell survival, we first focused on STAT signaling since we previously found that SHP2 silencing activates this pathway in endothelial cells (Salvucci *et al*, 2015). We now observed that SHP099 activates STAT3 in HUVEC (5 μM) and BMEC (20 μM), but not in the melanoma B16F10 cell line (20 μM) (Fig 3A and B). At these concentrations, SHP099 did not activate STAT1 or STAT5 and did not significantly alter AKT activity in endothelial and B16F10 cells (Appendix Fig S2A–E).

SHP2 is required for optimal activation of the ERK-MAP kinase pathway that sustains cell survival and proliferation in many cell types (Chan *et al*, 2008), including endothelial cells (Mavria *et al*, 2006; Chen *et al*, 2016). We found that SHP099 inhibits ERK activity in HUVEC, BMEC, and B16F10 cells (Fig 3A and B). Noteworthy, SHP099 reduces ERK activity, without activating STAT3 in B16F10 cells (Fig 3A and B). In addition, SHP099 activates STAT3 and inhibits ERK activity in endothelial cells beginning at similar early time points, raising the possibility that SHP2 may directly regulate these two pathways rather than change in one pathway be compensatory to the other.

Next, we evaluated the effects of STAT3 activation and ERK1/2 inactivation on endothelial cell proliferation. The JAK kinases inhibitor tofacitinib (Tofa, 50 nM) reduced endogenous p-STAT3 activity and enhanced the proliferation of HUVEC and BMEC compared to control (Fig 3C and D). In addition, the ERK1/2 inhibitor PD098059 (PD0980, 10 μM) reduced endogenous p-ERK1/2 activity and inhibited the proliferation of HUVEC and BMEC compared to control (Fig 3E and F). Furthermore, Tofa increased and PD0980 diminished BMEC viability (Appendix Fig S2F).

In a panel of cancer cell lines (Appendix Fig S2G), SHP099 reduced the proliferation of all but two cell lines (colon carcinoma RKO and MC38). In the growth-inhibited cell lines, SHP099 enhanced STAT3 and reduced ERK1/2 activity. However, in the RKO and MC38 lines, SHP099 neither reduced cell proliferation nor altered STAT3 and ERK1/2 activity. Since in B16F10 cells, SHP099 had no effect on cell proliferation and STAT3 activity, but reduced ERK1/2 activity (Fig 3A and B), we tested the effects of two well-established MAPK/ERK pathway inhibitors in B16F10 cells. AZD6244 and trametinib dose-dependently inhibited B16F10 cell proliferation (Fig 3G), reduced ERK1/2 activity, and additionally activated STAT3 (Fig 3H). MEK inhibitors were previously reported to induce "compensatory" STAT3 activation in other cancer cells (Vultur *et al*, 2014). Since SHP099, differently from MEK inhibitors, failed to induce STAT3 activation in B16F10 cells (Fig 3A), these

results raise the possibility that STAT3 activation may be a critical contributor to growth inhibition in B16F10 tumor cells. Importantly, these results indicate that SHP099 reduces endothelial cell proliferation and viability associated with STAT3 activation and ERK1/2 inhibition, suggesting that SHP099 may represent a previously unidentified anti-vascular agent.

## SHP099 promotes tumor vessel regression and reduces tumor growth

We selected the syngeneic B16F10-melanoma mouse model to evaluate the anti-vascular effects of SHP099. In vitro, B16F10 cells are more resistant to SHP099 growth inhibition than primary HUVEC and HDMEC (Fig 2E), and the likelihood of SHP099 having a direct anti-tumor effect appeared low since SHP099 transiently yielded maximal free plasma concentrations > 20 μM at the effective oral dose of 100 mg/kg (Chen *et al*, 2016). Also, the SHP099 target, SHP2, is phosphorylated in the CD31[+] vascular endothelial cells of B16F10 tumors whereas the CD31[−] tumor cells are generally p-SHP2[−] (Fig 1I).

We established subcutaneous B16F10 tumors in syngeneic mice. Groups of mice (7–11 mice/group) bearing similar size (range 30–50 mm³) tumors were treated daily with oral SHP099 (75–100 mg/kg) or vehicle only; the experiment was terminated when any tumor reached the maximum diameter of 20 mm in any direction. In three independent experiments, SHP099 reproducibly reduced B16F10 tumor growth and weight compared to control (Fig 4A–C), without causing loss of body weight (Figs EV3A and 3B) or evidence of other toxicity. This anti-tumor effect of SHP099 was associated with vascular leakage, as the FITC-dextran tracer (2,000,000 kDa) was frequently found outside the vessel wall in the tumor parenchyma where auto-fluorescent erythrocytes were also found (Figs 4D and EV3C). Type IV collagen immunostaining to visualize vascular basement membranes showed scattered remnants of vessels ("sleeves" (Inai *et al*, 2004)) that lacked endothelial CD31 coverage in SHP099-treated but generally not in control tumors (Figs 4E–G and EV3D). Tumor vascularity was also reduced (Fig 4F). Further supporting vessel involution, cleaved-caspase-3-positive endothelial cells decorated the wall of tumor vessels from the SHP099-treated more frequently than in control mice (Fig 4H and I). The Ki67[+] cells were reduced in SHP099-treated tumors compared to controls, indicative of reduced tumor cell proliferation (Fig 4J and K). In addition, tumor tissue hypoxia (Fig 4L), number of cleaved caspase-3[+] cells, particularly proximal to the tumor vessels (Figs EV3E and 3F), and tumor necrosis (Fig 4M and N) were increased in SHP099-treated tumors compared to controls. F4/80[+] infiltrates were similarly detected in control and SHP099-treated B16F10 tumors (Fig EV3G).

The prominent vascular pathology displayed by SHP099-treated tumors suggested a vascular basis of the anti-tumor effect of SHP099. The CD31[+] vasculature in liver, lung, heart, intestine, brain, and muscle appeared normal and was normally perfused in tumor-bearing mice after SHP099 treatment, and the tissues appeared normal (Appendix Fig S3). Thus, the selectivity of SHP099 for the tumor vasculature may reflect differences between quiescent vessels in adult tissues and remodeling vessels of tumors.

SHP099 may elicit anti-tumor immune cell responses by attenuating signaling from immune-inhibitory receptors, such as PD-1 (Chen

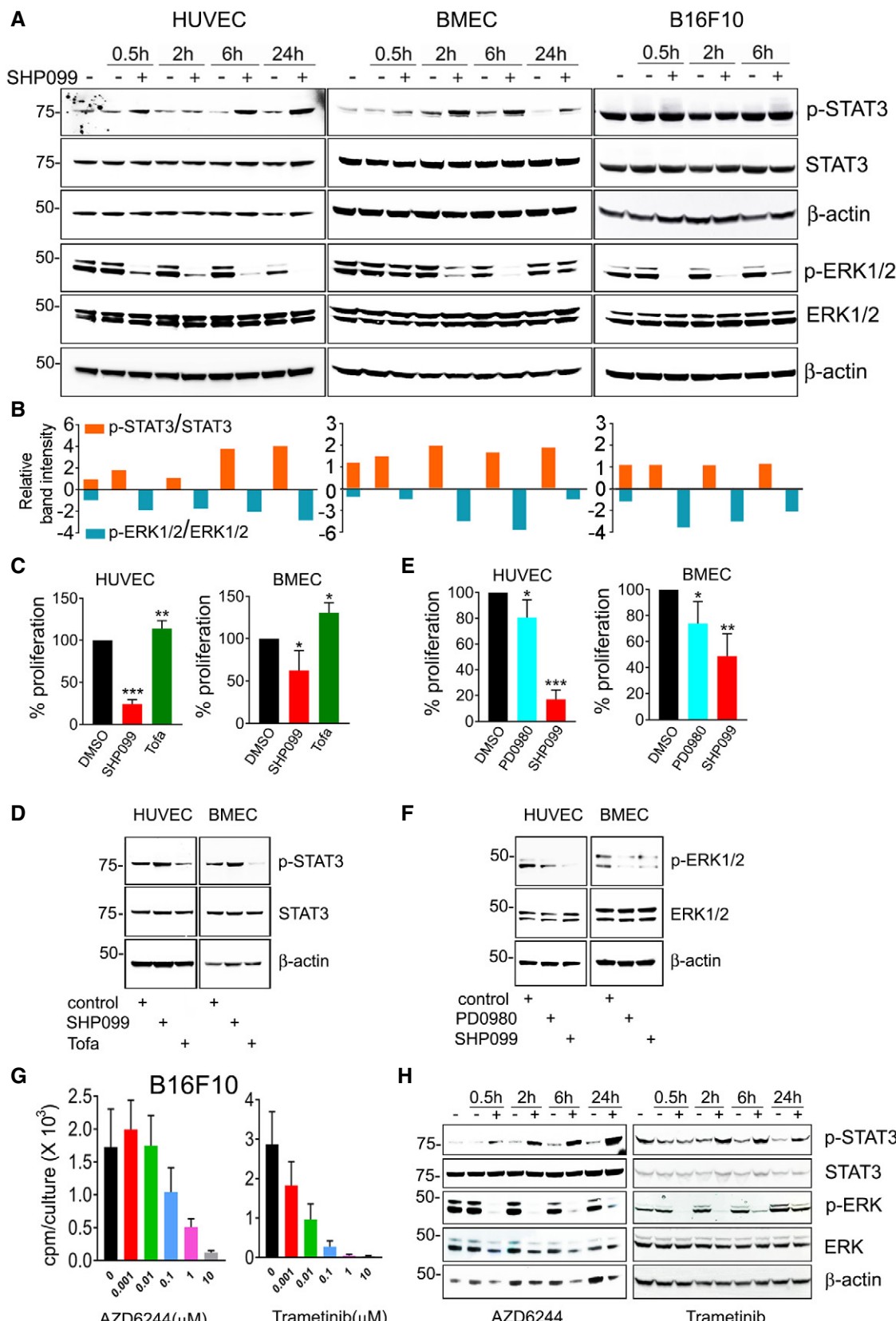

Figure 3.

**Figure 3.  SHP099 reduces ERK1/2 and increases STAT3 phosphorylation in endothelial cells.**

A    Kinetic SHP099 modulation of p-STAT3 (Tyr$^{705}$) and p-ERK1/2 (Thr$^{202}$/Tyr$^{204}$) in endothelial and B16F10 cells; SHP099: 5 μM HUVEC; 20 μM BMEC and B16F10; representative immunoblotting of 3–5 experiments; h: hours.

B    Quantification of band intensities in panel (a); p-STAT3/STAT3 and p-ERK1/2/ERK1/2.

C–F   (C,D) The JAK inhibitor Tofacitinib (Tofa, 50 nM) enhances the proliferation of HUVEC and BMEC compared to control (C) and reduces endogenous p-STAT3 levels in HUVEC and BMEC compared to control (D) after 72-h culture. SHP099 (5 μM HUVEC; 20 μM BMEC) reduces cell proliferation and increases p-STAT3 in HUVEC and BMEC compared to control; error bars: ± SD; (E,F) The MAP kinases inhibitor PD098059 (PD0980, 10 μM) and SHP099 (5 μM HUVEC; 20 μM BMEC) inhibit HUVEC and BMEC proliferation compared to control (E) and reduce p-ERK1/2 levels by immunoblotting after 72-h culture (F). Results of proliferation in panels C and E (% of control) from triplicate cultures are representative of 3–4 experiments; *P < 0.05, **P < 0.01 ***P < 0.001; two-tailed Student's t-test; error bars: ± SD. Results of immunoblotting in panels D and F are representative of three independent blots.

G, H   The ERK1/2 inhibitors AZD6244 and tamatinib dose-dependently reduce B16F10 cell proliferation (G); representative experiment of three independent experiments; the results reflect the means of five replicate/condition; error bars: ± SD; and levels of endogenous p-STAT3 and p-ERK in B16F10 cells (representative of three blots) (H).

Source data are available online for this figure.

et al, 2016; Hui et al, 2017; Quintana et al, 2020). Therefore, we looked for a potential contribution of anti-tumor immunity to the anti-tumor activity of SHP099 in B16F10 tumors. Given that T and NK cells promote the effectiveness of checkpoint blockade (Hsu et al, 2018; Ngwa et al, 2019), we utilized the T- and NK-cell deficient NSG (NOD scid gamma) mice. Due to the rapid growth of B16F10 tumors in NSG mice, treatment was initiated 72 h after tumor cell injection. SHP099 showed similar anti-tumor (Fig 5A) and anti-vascular (Fig 5B and C) activity in immunodeficient NSG and immunocompetent B6 mice. Also, SHP099 did not substantially change the number of F4/80$^{+}$ tumor-infiltrating macrophages in these NSG mice (Figs EV4A and 4B). These results suggest that SHP099 activation of anti-tumor immunity is not critical in the current system.

SHP099 did not reduce the proliferation or viability of B16F10 cells in vitro. However, it remained possible that tumor cells acquired responsiveness to SHP099 in vivo, which could contribute to the anti-tumor activity of SHP099. To address this possibility, we utilized SHP2-depleted B16F10 cells (Fig 2F) and injected SHP2-depleted (shSHP2#2) B16F10 cells into syngeneic mice to generate tumors along with vector-control B16F10 cells. SHP099 (100 mg/kg orally/daily) reduced the growth of SHP2-depleted and non-depleted B16 tumors to a similar degree (Fig 5D and E). We confirmed that tumor SHP2 was markedly reduced at completion of the experiment (Fig 5F). The vascular phenotype of increased vascular leakage (Fig 5G) and reduced vascularization (Fig 5H) was similar in SHP2-depleted and non-depleted tumors treated with SHP099. These results demonstrate that SHP099 is a previously unrecognized anti-vascular drug that damages the tumor endothelium and likely reduces tumor growth through a vascular mechanism.

Given the extensive vascular pathology in SHP099-treated tumors, we examined whether increased vascular permeability leads to increased metastatic dissemination. To address this possibility, we injected B16F10 cells (0.5 × 10$^{6}$) in the footpad of syngeneic mice and looked for metastasis in the draining popliteal lymph nodes and the lungs. Beginning on day 4 after B16F10 cell inoculation, groups of mice were treated with SHP099 (n = 20; 100 mg/kg orally/daily) or with vehicle only (n = 20) (Fig 5I). All mice were euthanized when the primary tumor in the footpad reached a maximum of 10mm in any direction in any mouse (day 21 from injection of the B16F10 cells). The SHP099-treated primary tumors were smaller than the controls (Fig 5J). The popliteal lymph nodes, ipsilateral to the site of tumor cell injection, from SHP099-treated mice

(18/20 recovered) were smaller than those (n = 19/20 recovered) from the control mice (Fig 5K). Visual inspection detected metastatic melanoma in a lower percentage of SHP099-treated mice than in control (Figs 5L and EV4C). Microscopic evaluation of H&E-stained popliteal lymph nodes (without knowledge of treatment status) revealed the presence of micro-metastasis in 70% of SHP099-treated mice and 100% of controls (Fig 5M). The extent of metastatic infiltration in the control group was generally greater than in the SHP099-treated mice (Fig 5N). None of the lungs had visible metastasis. One of the lungs from a control mouse (n = 10 controls and n = 10 SHP099-treated examined) had microscopic evidence of single metastatic infiltration. These results provide evidence that SHP099 treatment does not promote dissemination of B16F10 melanoma cells from the primary tumor to the draining lymph node.

## SHP2 and Ang/TIE2 inhibitors cooperatively inhibit tumor vascularization and tumor growth

Despite its effectiveness, SHP099 did not induce a complete involution of the tumor vasculature and did not eradicate B16F10 tumors (Fig 4A–C). To target the vessels that persisted after SHP099 treatment, we sought to inhibit Ang2/TIE2-induced tumor angiogenesis that is mediated by the PI3K/AKT pathway (Augustin et al, 2009; Saharinen et al, 2017). SHP099 does not block PI3K/AKT signaling in endothelial cells (Appendix Fig S2C and D) and other cell types (Appendix Fig S2E) (Fedele et al, 2018). This pathway is active in the vasculature of B16F10 tumors as TIE2 is broadly phosphorylated in the endothelial cells of B16F10 tumors (Fig 1I) and TIE2 phosphorylation persists in the tumor vasculature of mice treated with SHP099 (Fig EV4D).

AMG386, a peptibody that inhibits Ang1/Ang2-TIE2 binding and impairs tumor angiogenesis (Coxon et al, 2010; Scholz et al, 2016) significantly reduced the growth of established B16F10 tumors compared to control (Fig 6A and B). The anti-tumor effect of AMG386 (5.6 mg/kg s.c.; twice/week) was similar to that of SHP099 (P = 0.93, exp.1, panel A; P = 0.50 exp. 2, panel B). Together, SHP099 and AMG386 cooperatively reduced tumor growth and weight (Fig 6A; SHP099 vs SHP099+ANG386: P = 0.006 exp. 1, 0.07 exp. 2; AMG386 vs SHP099+AMG386: P = 0.0002 exp. 1, P = 0.049 exp. 2); one of the tumors underwent full regression. Like SHP099, AMG386 minimally affected the proliferation of B16F10 cells (Fig EV4E), which do not express TIE2 (Nair et al,

2003). Mouse body weight was similar after treatment with AMG386 alone or with SHP099 (Fig EV3B).

As a single agent, AMG386 reduced tumor vascular p-TIE2 (Fig 6C and D) and p-EphrinB (Fig 6E and F); these results are consistent with Ang2 activation of p-TIE2 and p-EphrinB in primary endothelial cells

(Fig 1H). In addition, AMG386 compromised the integrity of tumor vessels as reflected by localized FITC-dextran extravasation into the tumor parenchyma (Fig 6G) and the presence of cleaved caspase-3[+]/CD31[+] cells lining blood vessels (Figs 6H and EV4F). Compared to control, AMG386 reduced the CD31[+] vessel area (Fig 6I) and

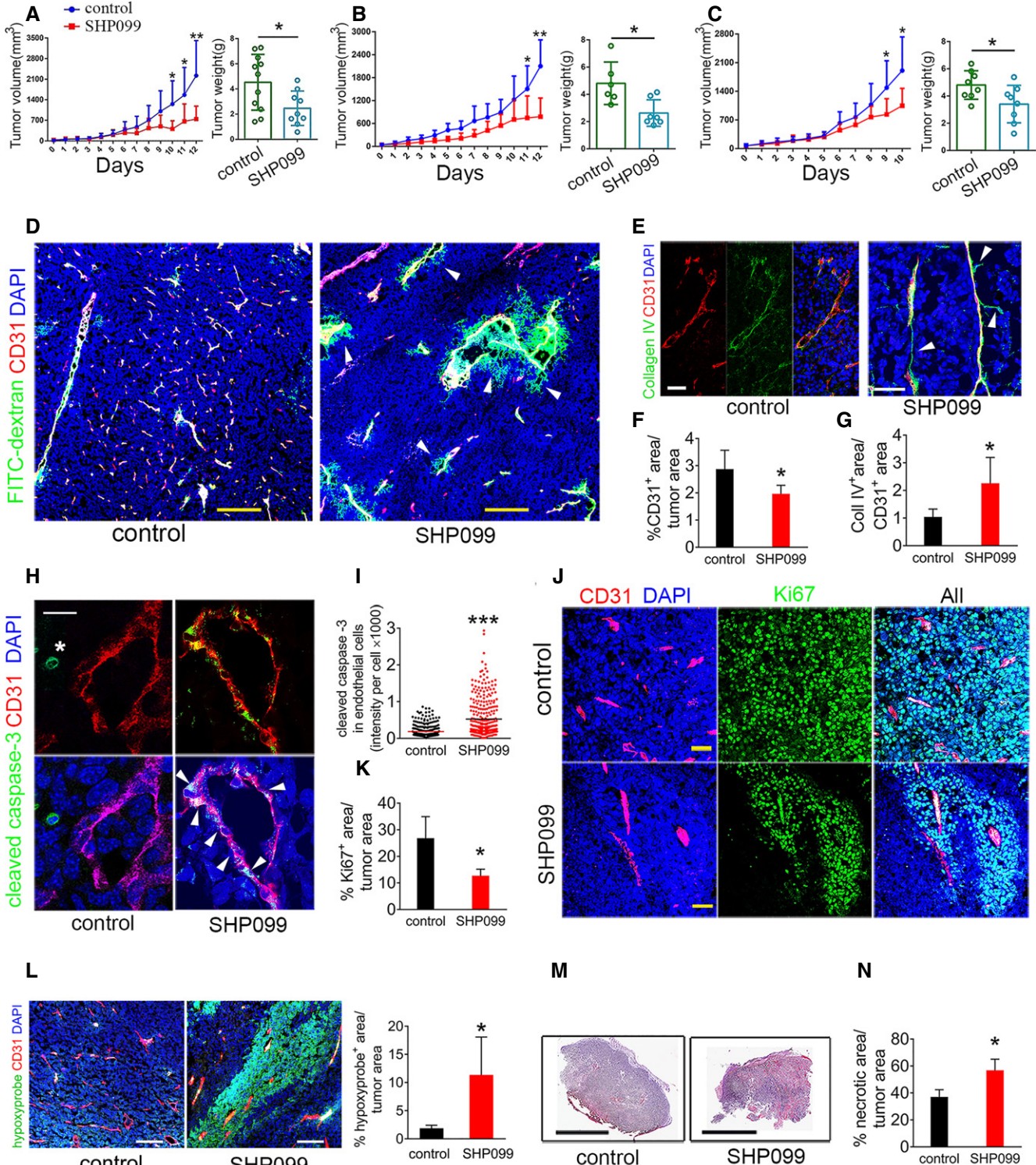

Figure 4.

**Figure 4. Vascular effects of SHP099 and inhibition of B16F10 tumor growth.**

A–C SHP099 (A, 75 mg/kg; B, C: 100 mg/kg) reduces B16F10 tumor growth in syngeneic mice. Panels depict tumor growth from initiation of treatment to endpoint (left) and tumor weight (right) at endpoint; no. mice/group: 6–11 control; 7–9 SHP099.

D Extravasation of FITC-dextran (green, pointed by arrowheads) in tumors treated with SHP099. Vessels identified by CD31 (red); DAPI (blue) identifies cell nuclei. Confocal images (scale bar: 200 μm). Representative images from 3 tumors.

E "Vascular sleeves" (Collagen IV$^+$ CD31$^−$) identified in a representative SHP099-treated tumor. Representative confocal images from 6 tumors (scale bar: 50 μm).

F Reduced CD31$^+$ vascular area in SHP099-treated tumors compared to controls ($n = 6$/group); quantification by ImageJ.

G The ratio of Collagen IV$^+$/CD31$^+$ area is increased in SHP099-treated tumors compared to controls ($n = 6$/group); quantification by ImageJ; Coll: collagen.

H Cleaved caspase-3 detection in a CD31$^+$ vessel (representative of 6 SHP099-treated tumors); arrowheads point to cleaved caspase-3$^+$CD31$^+$ cells. Cleaved caspase-3$^+$ tumor cell is marked by *. Confocal images (scale bar: 10 μm).

I Increased cleaved caspase-3 fluorescence intensity in CD31$^+$ cells from SHP099-treated tumors compared to controls ($n = 3$/group); at least 100 cells counted/sample.

J Ki67$^+$ proliferating cells in representative of 3 SHP099-treated and 3 control tumors; CD31 identifies the vessels; DAPI identifies cell nuclei. Confocal images (scale bar: 50 μm).

K Quantification of proliferating cells within SHP099 and control ($n = 4$/group) tumors by ImageJ.

L Hypoxic tumor identified Hypoxyprobe (green) representative of 3 SHP099-treated tumors; confocal images (scale bar: 100 μm), left; quantification of hypoxic areas in control ($n = 3$) and SHP099-treated ($n = 3$) tumors by ImageJ.

M Representative control (of 5) and SHP099-treated (of 5) B16F10 tumor sections through the maximum diameter; H&E staining (scale bar: 10mm).

N Cleaved caspase-3$^+$ fluorescence intensity in SHP099 and control tumors ($n = 3$/group).

Data information: Error bars: ± SD; P values from two-tailed Student's t-test; *P < 0.05, **P < 0.01 and ***P < 0.001.

promoted formation of vascular "sleeves" (Figs 6J and EV4G). AMG386 reduced viable and proliferative tumor burden (Fig 6K–N).

Dual treatment with SHP099 and AMG386 resulted in a remarkable reduction of tumor vascularity (Fig 6I) and viable/proliferative tumor burden (Fig 6K–N). Some of the residual tumor vessels appeared damaged and leaky (Figs 6G and EV4F) and not viable based on the presence of cleaved caspase 3$^+$/CD31$^+$ endothelial cells lining vessels (Figs 6H and EV4F) and detection of vascular sleeves (Figs 6J and EV4G). Unlike the anti-vascular effects of SHP099+AMG386 in the tumors, the vasculature in the liver, lung, heart, intestine, brain, and muscle of mice treated with AMG386+SHP099 appeared normal and normally perfused (Appendix Fig S4).

We examined the activity of intracellular signaling targets of SHP099 and AMG386 in the tumor vasculature. Nuclear p-STAT3 was detected at significantly higher levels in the tumor vasculature of mice treated with SHP099, AMG386, or SHP099+AMG386 compared to control (Fig 7A and B), but the increase in p-STAT3 induced by AMG386 alone or with SHP099, was quantitatively lower than induced by SHP099 alone ($P < 0.001$ both comparisons). Also, nuclear/active p-ERK1/2 was significantly reduced in the tumor vasculature of mice treated with SHP099, AMG386, or SHP099+AMG386 compared to control (Fig 7C and D). The reduction of p-ERK1/2 from AMG386 treatment was significantly lower than from SHP099 treatment ($P < 0.001$), whereas the combination of SHP099 and AMG386 was more effective than SHP099 alone ($P < 0.001$) (Fig 7D). These results, showing that SHP099 activates p-STAT3 and reduces p-ERK1/2 in tumor endothelial cells of mice, mirror the effects of SHP099 in endothelial cells *in vitro* (Fig 3A and B). Additionally, the results showing that AMG386 reduces p-ERK1/2 and activates STAT3 in tumor endothelial cells are consistent with the reduced activity of EphrinB in tumor endothelial cells of mice treated with AMG386 (Fig 6C). It was previously known that active EphrinB2 not only plays an important role in restraining STAT signaling in endothelial cells (Salvucci *et al*, 2015) but also promotes VEGF/VEGFR internalization/signaling resulting in ERK1/2 activation (Sawamiphak *et al*, 2010; Wang *et al*, 2010).

As an indicator of Ang1/2-TIE2/p-AKT signaling in the tumor vasculature, we examined the subcellular localization of the transcription factor Forkhead box protein O1 (FOXO1) (Daly *et al*, 2006). Active AKT leads to FOXO1 translocation from the nucleus to the cytoplasm where it is rapidly degraded (Aoki *et al*, 2004; Saharinen *et al*, 2017), predicting that reduced AKT activity would lead to accumulation of FOXO1. By immunofluorescence, FOXO1 was mostly restricted to the DAPI$^+$ nuclei of CD31$^+$ tumor endothelial cells in control and treated mice (Fig 7E), attributable to the rapid degradation of cytoplasmic FOXO1. Quantitatively, the tumor vasculature of AMG386-only or AMG386+SHP099-treated mice was significantly enriched with FOXO1 compared to control mice, whereas the tumor vasculature of SHP099-treated mice was not ($P = 0.38$) (Fig 7F).

In additional experiments, we evaluated the effects of treatment with SHP099 and AMG386 on MC38 colon cancer established in syngeneic mice. The MC38 cells, such as B16F10 cells, are not growth inhibited by 5–20 μM SHP099 in vitro (Appendix Fig S2G), thus providing an additional mouse tumor type where to evaluate the anti-vascular activity of SHP099. MC38 tumors resemble B16F10 tumors in showing that EphrinB, TIE2, and SHP2 are active in the tumor vasculature, and most tumor cells are Ang2$^+$ (Fig EV5A).

Individually, SHP099 (100 mg/kg, daily gavage) and AMG386 (5.6 mg/kg s.c.; twice/week) reduced MC38 tumor growth and tumor weight (SHP099 vs control $P < 0.001$; AMG386 vs control $P < 0.05$; SHP099 vs AMG386, $P = 0.271$; Fig 8A and B). Together, SHP099 and AMG386 cooperatively reduced tumor growth and weight (SHP099 vs SHP099+AMG386: $P = 0.0008$; AMG386 vs SHP099+AMG386: $P < 0.0001$) (Fig 8A and B). This anti-tumor activity of SHP099 and AMG386, individually and together, was associated with vascular leakage of the FITC-dextran tracer in the tumor parenchyma (Fig 8C), an increase in the number of "vascular sleeves" (Figs 8D and EV5B) and an increase of cleaved-caspase-3$^+$ endothelial cells (Figs 8E and EV5B) compared to untreated controls. This vascular pathology was associated with reduced tumor vascularity (Fig 8F) and reduced tumor cell proliferation in all treatment groups compared to control (Fig 8G and H). Tumor tissue necrosis was increased in all treatment groups compared to control, particularly with the combination of SHP099 plus AMG386 (Fig 8I).

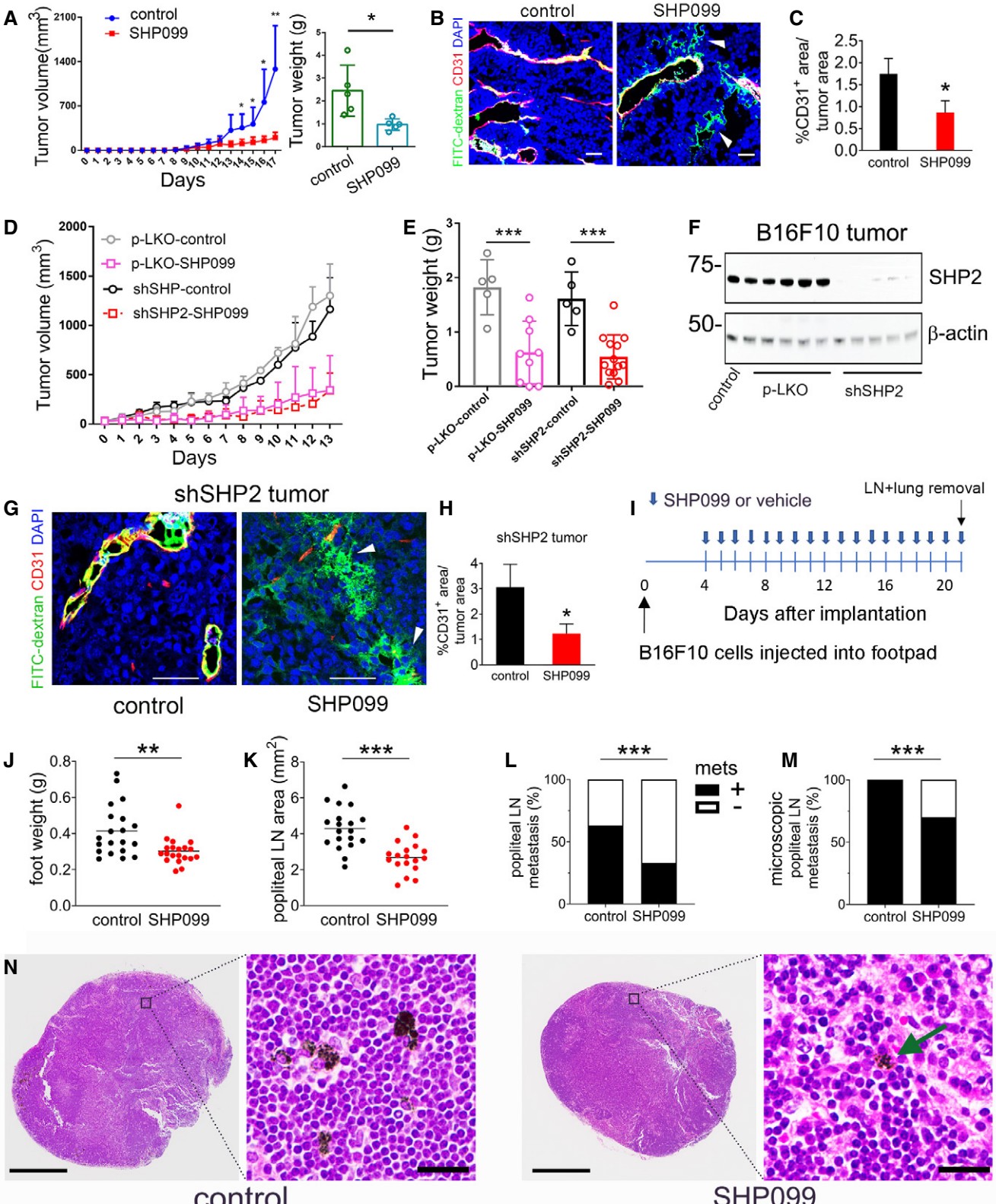

**Figure 5.**

**Figure 5. Features of SHP099 tumor growth inhibition.**

A  Anti-tumor activity of SHP099 against B16F10 tumors established s.c. in NOD scid (NSG) mice ($n = 5$/group); $P$ values from two-tailed Student's $t$-test; *$P < 0.05$, **$P < 0.01$. Error bars: ± SD.

B  Extravasation of FITC-dextran (green) in tumors established in NSG mice treated with SHP099; scale bar (50 μm). Representative images from 3 mice/group.

C  Reduced CD31+ vascular area in SHP099-treated tumors compared to controls ($n = 5$/group); ImageJ quantification. $P$ values from two-tailed Student's $t$-test; *$P < 0.05$. Error bars: ± SD.

D, E  SHP099 reduces growth (D) and weight (E) of control (pLKO vector) and SHP2-depleted (shSHP2) B16F10 tumors compared to untreated controls: pLKO untreated ($n = 5$), pLKO treated with SHP099 ($n = 9$), shSHP2 untreated ($n = 5$), and shSHP2 treated with SHP099 ($n = 13$). $P$ values by analysis of variance with Dunnett's multiple comparison test; ***$P < 0.001$.

F  Persistent depletion of SHP2 in tumors induced by shSHP2-B16F10 cells ($n = 5$) compared to control tumors induced by pLKO-B16F10 ($n = 5$); tumors removed from mice 18 days after cell inoculation.

G  SHP099 induces extravasation of FITC-dextran (green) in tumors from inoculation of shSHP2-B16F10 (representative image from 3 tumors); scale bar: 50 μm.

H  Reduced CD31+ vascular area in SHP099-treated tumors from s.c. inoculation of shSHP2-B16F10 cells compared to controls from pLKO-B16F10 cells ($n = 5$/group); ImageJ quantification. Error bars: ± SD; $P$ values from two-tailed Student's $t$-test; *$P < 0.05$.

I  Schematic of experiment.

J, K  After footpad inoculation of B16F10 cells, mice were treated orally with SHP099 ($n = 20$) or vehicle only ($n = 20$). Tumor size (J) was estimated from foot weight (g); size of popliteal lymph nodes (K) was measured (mm$^2$) by ImageJ software; Student's $t$-test; **$P < 0.01$, ***$P < 0.001$.

L, M  Proportion of mice in the SHP099 and control groups with macroscopic (L) and microscopic (M) evidence of metastases; Fisher's exact test; ***$P < 0.001$.

N  Representative popliteal lymph nodes from control and SHP099-treated mice with microscopic evidence of metastases; H&E stain; scale bars 500 μm (low magnification) 20 μm (high magnification).

As expected, AMG386 reduced significantly TIE2 and EphrinB phosphorylation in the vasculature of MC38 tumors (Fig EV5C–E). Also, SHP099 and AMG386, individually or together increased significantly p-STAT3 and p-ERK1/2 levels in tumor CD31+ endothelial cells (Figs 8J and K, and EV5F). Additionally, AMG386 alone or with SHP099 increased substantially FOXO1 in tumor endothelial cells (Figs 8L and EV5F). However, SHP099 did not significantly alter tumor infiltration by CD3+ lymphocytes and F4/80+ macrophages alone (Figs EV5G and 5H).

Together, these results support a model (Fig 8M) by which SHP099 and AMG386 modulate signaling cascades in endothelial cells reducing tumor vascularization.

## Discussion

In many cancer types, SHP2 is hyperactive due to mutations of the Ptpn11 gene, which encodes SHP2, or to aberrant signaling from protein tyrosine kinases, and functions as a proto-oncogene (Ran et al, 2016). Hence, SHP2 is a therapeutic target in cancer. The specific SHP2 inhibitor, SHP099, effectively suppressed the growth of receptor–tyrosine–kinase-driven cancers in mouse xenograft models (Chen et al, 2016) and prevented cancer resistance to inhibitors of MEK (Fedele et al, 2018; Mainardi et al, 2018; Wong et al, 2018) and anaplastic lymphoma kinase (ALK) (Dardaei et al, 2018).

Here, using chemical inhibition of SHP2 and genetic approaches to deplete cellular SHP2, we show that the SHP2 phosphatase plays a key role as a mediator of tumor vessel persistence and identify SHP2 as a new pharmacological target for the successful anti-vascular therapy of cancer. SHP2 is a ubiquitously expressed intra-cellular tyrosine phosphatase that promotes ERK signaling induced by growth factors, cytokines, and hormones and supports cell survival and proliferation (Chan et al, 2008; Ran et al, 2016). SHP099 promoted the involution of the remodeling vasculature of tumors and inhibited tumor neovascularization while having no detectable direct effect on the tumor cells or the resting vasculature of normal tissues, which substantially differs from the tumor vasculature (Potente et al, 2011). A recent study noted that

vascularization was generally reduced in tumors treated with SHP099, but it remained unclear whether this was attributable to a primary effect on the tumor vessels or an indirect consequence of tumor cell targeting (Fedele et al, 2018). In our study, the anti-vascular activity of SHP099 is unlikely a consequence of an effect on the tumor cells, since the genetic depletion of SHP2 from the B16F10 tumor cells had no impact on the anti-vascular and anti-tumor activities of SHP099. Thus, B16F10 cancer model differs from the mutant KRAS$^{G12C}$ MIAPaCA-2 cancer model, in which the effectiveness of SHP099 was reported to be cancer cell intrinsic and the effects of SHP099 were recapitulated by the genetic depletion of SHP2 in the cancer cells (Hao et al, 2019). A contribution of T/NK-induced anti-cancer immunity appears unlikely in the B16F10 model, since SHP099 exerted similar anti-vascular and anti-tumor activities in immunocompetent and immunodeficient NSG mice. Mechanistically, SHP099 inhibited ERK1/2 phosphorylation and activated STAT3 phosphorylation in endothelial cells from culture and the tumor vasculature. This signaling regulation likely mediates the anti-vascular effects of SHP099, since ERK1/2 signaling is critical to endothelial cell sprouting and angiogenesis (Mavria et al, 2006) and STAT signaling promotes endothelial cell death and vessel involution (Kreuzaler et al, 2011; Salvucci et al, 2015). Thus, the current results spotlight a spectrum of SHP2 functions in the vascular endothelium and spur clinical development of SHP099 and similar inhibitors as novel anti-vascular agents.

Genetic experiments in mice showed that SHP2 is required for the formation of a hierarchically organized vessel network in the mouse yolk sac, linking SHP2 to regulation of post-angiogenic/vasculogenic remodeling events (Saxton et al, 1997). Similarly, the knockout of SHP2 in endothelial cells caused embryonic hemorrhage and death attributed to disruption of endothelial cell junctions (Zhang et al, 2019). Experiments in vitro suggested that SHP2 regulates endothelial adhesive functions and migration stemming from SHP2 binding to the intracellular domain of endothelial platelet-derived growth factor receptor-b (PDGFR-b), VE-cadherin, and PECAM-1 (Qi et al, 1999; Ukropec et al, 2000). The knockdown of SHP2 transiently reduced FGF2+VEGF-dependent endothelial cell viability in vitro and ex vivo (Mannell

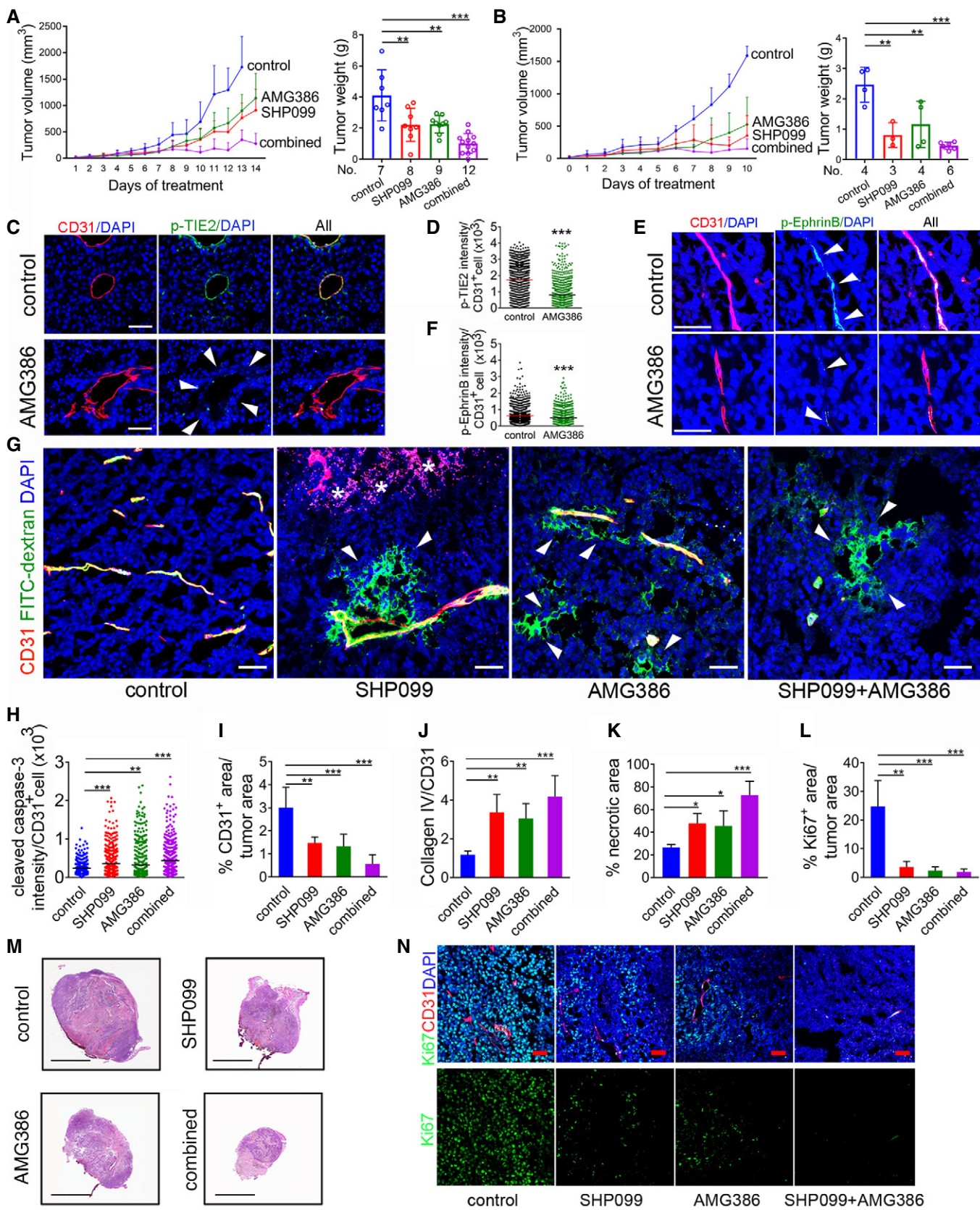

**Figure 6.**

**Figure 6. SHP099 and AMG386 cooperatively reduce B16F10 tumor growth.**

A, B    AMG386 (5.6 mg/kg s.c.; twice/week) and SHP099 (a: 200 mg/kg; b: 100 mg/kg orally/daily) were administered individually or together to B16F10 tumor-bearing
        mice (A: control $n = 7$; SHP099: $n = 8$; AMG386: $n = 8$; SHP099+AMG386 $n = 11$; B: control, SHP099 and AMG386: $n = 4$; SHP099+AMG386: $n = 8$). Panels depict
        tumor growth from initiation of treatment to endpoint (left) and tumor weight (right) at endpoint; error bars $\pm$ SD; $P$ values by analysis of variance with Dunnett's
        multiple comparison test; **$P < 0.01$, ***$P < 0.001$.

C, D    p-TIE2 in tumor vessels from control and AMG386-treated mice; representative confocal images; arrowheads point to CD31[+]/p-TIE2[−] endothelial cells (scale
        bars:50 μm) (C), and quantification (D) of p-TIE-2/CD31[+] in control ($n = 3$) and AMG386-treated ($n = 3$) tumors (1,200 CD31[+] cells/group). $P$ values from two-tailed
        Student's $t$-test; ***$P < 0.001$.

E, F    p-EphrinB (E) in representative tumor vessels from control ($n = 3$) and AMG386-treated ($n = 3$) tumors; confocal images (E), (scale bar: 50 μm) and quantification of
        p-EphrinB/CD31[+] (F); $P$ values from two-tailed Student's $t$-test; ***$P < 0.001$.

G       Vessel perfusion visualized by FITC-dextran (green); CD31 (red) identifies the endothelium; DAPI (blue) identifies the nuclei; representative tumor confocal images
        (scale bar: 50 μm) of 3 mice/group; arrowheads point to FITC-dextran extravasation; asterisks to extravasated erythrocytes.

H       Quantification of cleaved caspase-3 in the tumor vasculature (CD31[+] cells); $n = 3$ tumors/group; number of CD31[+] cells counted: control ($n = 311$); SHP099
        ($n = 425$); AMG386 ($n = 451$); combined SHP099+AMG386 ($n = 484$). Error bars: $\pm$ SD; $P$ values by analysis of variance with Dunnett's multiple comparison test;
        **$P < 0.01$, ***$P < 0.001$.

I       Quantification of tumor vascular area; $n = 5$/tumors group. Error bars: $\pm$ SD; $P$ values by analysis of variance with Dunnett's multiple comparison test; **$P < 0.01$,
        ***$P < 0.001$.

J       Quantification of "vascular sleeves" (CD31[−] Collagen IV[+]) $n = 5$/tumors group. Error bars: $\pm$ SD; $P$ values by analysis of variance with Dunnett's multiple comparison
        test; **$P < 0.01$, ***$P < 0.001$.

K, L    (K) % necrotic tumor area ($n = 4$/tumors group) and (L) % proliferating tumor cells ($n = 3$/tumors group) in control and treated groups. Error bars: $\pm$ SD; $P$ values
        by analysis of variance with Dunnett's multiple comparison test; *$P < 0.05$, **$P < 0.01$, ***$P < 0.001$.

M       Representative control and treated ($n = 5$/group) B16F10 tumor sections through the maximum diameter; H&E staining; (scale bars: 10mm).

N       Proliferating Ki67[+] cells (green) in representative control and treated tumors ($n = 3$/group); CD31 identifies the vessels; DAPI identifies cell nuclei. Confocal images
        (scale bar: 50 μm).

Source data are available online for this figure.

*et al,* 2008), delayed vessel regeneration in a mouse skin wounding model (Rieck *et al,* 2019) and increased endothelial monolayer permeability (Zhang *et al,* 2019). Also, SHP2 promoted VEGF-dependent ERK1/2 signaling (Simons *et al,* 2016), consistent with the role of EphrinB2-SHP2 signaling as a key regulator of VEGFR2 internalization and ERK signaling (Sawamiphak *et al,* 2010; Wang *et al,* 2010). However, SHP2 was also identified as an inhibitor of VEGFR2 activity and internalization in cells grown on collagen I, but not vitronectin-coated surfaces (Mitola *et al,* 2006).

VEGF is a principal inducer of neovascularization and vascular permeability (Apte *et al,* 2019). Anti-VEGF/VEGFR2 therapies are effective at reducing angiogenesis and tumor growth in mouse tumor models and in patients with certain cancers (Apte *et al,* 2019). The PLCγ–ERK1/2 cascade, which mediates VEGF/VEGFR2 signaling from phosphorylated tyrosine 1173, plays a central role in VEGF-induced proliferation/angiogenic sprouting (Simons *et al,* 2016). By inhibiting ERK1/2 signaling, SHP099 is expected to reduce VEGF-induced sprouting angiogenesis. However, the context-dependent role of SHP2 as a negative regulator of VEGFR2 signaling and angiogenesis highlights the complexities of VEGF/VEGFR2 regulation, and the requirement for further investigation (Mitola *et al,* 2006).

SHP099 incompletely blocked B16F10 tumor neovascularization. We hypothesized that this was attributable to anti-VEGF cell resistance, previously linked to Ang2, FGF2, IL-6, IL-10, and other factors (Sitohy *et al,* 2011; Eichten *et al,* 2016; Apte *et al,* 2019). Here, we identified Ang2/TIE2 as a pro-angiogenic pathway active in B16F10 tumors, which secrete Ang2 (Abdul Pari *et al,* 2020) and found that Ang1/2-TIE2 neutralization enhances the efficacy of SHP099 as an anti-angiogenic and anti-cancer agent. Consistent with this, inhibition of Ang2 contributed to overcoming resistance to VEGF blockade (Scholz *et al,* 2016). Ang1/TIE2 signaling activates the PI3K/AKT cascade leading to inhibition of the transcription factor Forkhead box protein O1 (FOXO1) and reduced transcription of FOXO1 target genes (Saharinen *et al,* 2017). Ang2 generally functions as a TIE2 agonist in the context of cancer, presumably activating the PI3K/AKT cascade (Saharinen *et al,* 2017), a result we confirm here. Thus, cooperation between SHP099 and Ang1/2-TIE2 neutralization may derive from blocking non-overlapping pro-angiogenic cascades.

Endothelial cell death, vascular damage, and involution of the tumor vasculature emerge as dominant effects of SHP099, associated with loss of vascular integrity, blood extravasation, and tumor hypoxia. These pathologies are more accentuated than with anti-

**Figure 7. SHP099 and AMG386 modulate STAT3, ERK, and AKT signaling in the B16F10 tumor vasculature.**

A–D    Nuclear p-STAT3 (A,B) and p-ERK1/2 (C,D) in tumor CD31[+] endothelial cells of mice treated for 14 days with SHP099 (200 mg/kg), AMG386 (5.6 mg/kg) or the
        combination of SHP099 (200 mg/kg) + AMG386 (5.6 mg/kg). Representative (3 tumors/group) immunostaining (A,C) and quantification of staining intensity/CD31[+]
        cell (B, D); horizontal line: mean intensity/group; $n = 3$ tumors/group; number of CD31[+] cells counted in B: control ($n = 180$), SHP099 ($n = 308$), AMG386 ($n = 300$),
        combined SHP099+AMG386 ($n = 322$); number of CD31[+] cells counted in D: control ($n = 1,733$), SHP099 ($n = 821$), AMG386 ($n = 1,399$), combined SHP099+AMG386
        ($n = 783$). Scale bars: A, 50 μm; C, 10 μm.

E, F    FOXO1 detection in tumor CD31[+] endothelial cells after treatment with SHP099 (200 mg/kg), AMG386 (5.6 mg/kg) or the combination of SHP099 (200 mg/
        kg) + AMG386 (5.6 mg/kg). Representative (3 tumors/group) confocal images (E) and quantification ($n = 3$ tumors/group) (F) of FOXO1 fluorescence intensity/CD31[+]
        cell. Number of CD31[+] cells counted: control ($n = 2,285$), SHP099 ($n = 2,262$), AMG386 ($n = 2,300$), combined SHP099+AMG386 ($n = 2,361$).

Data information: $P$ values by analysis of variance with Dunnett's multiple comparison test; **$P < 0.01$, ***$P < 0.001$.

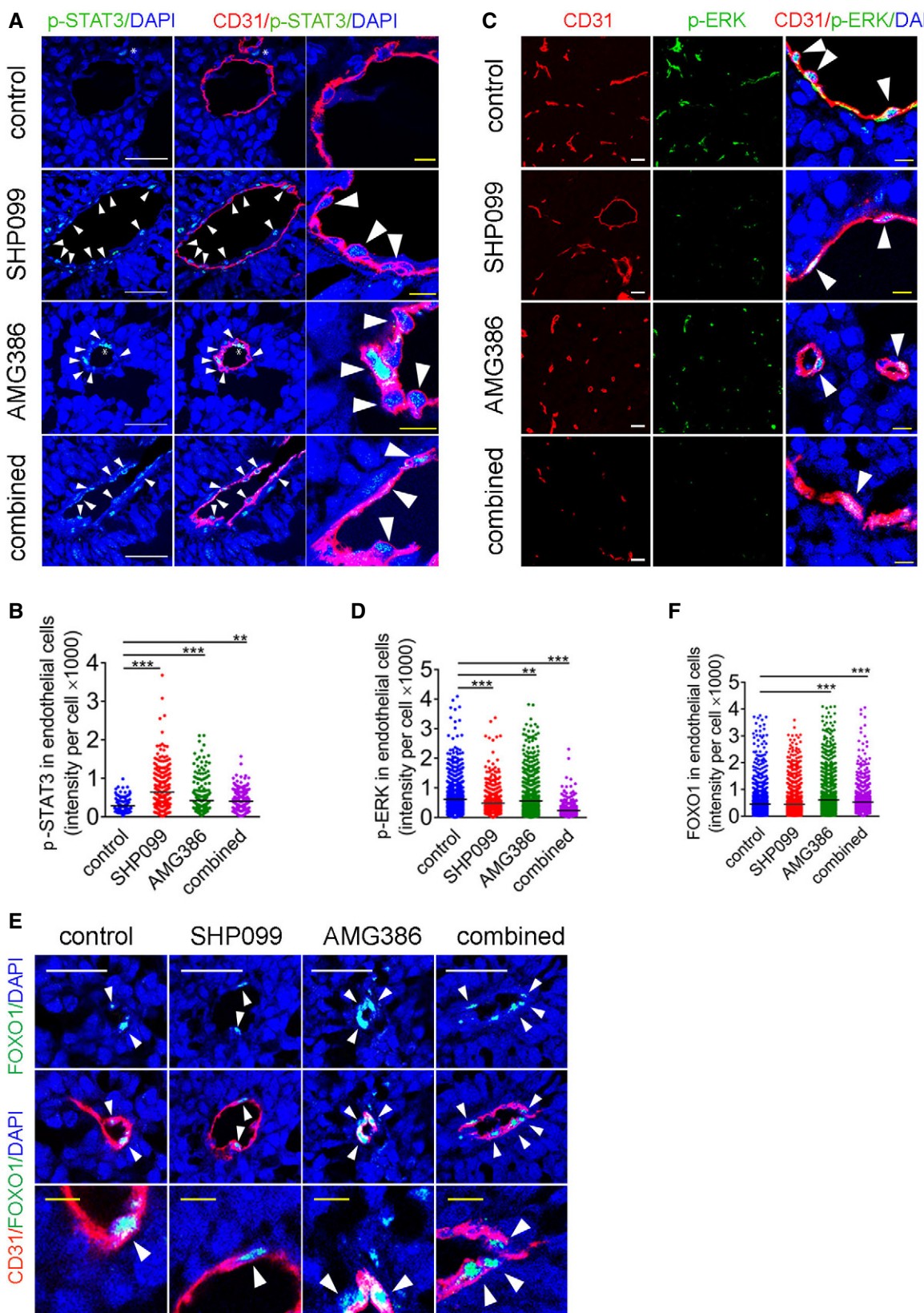

**Figure 7.**

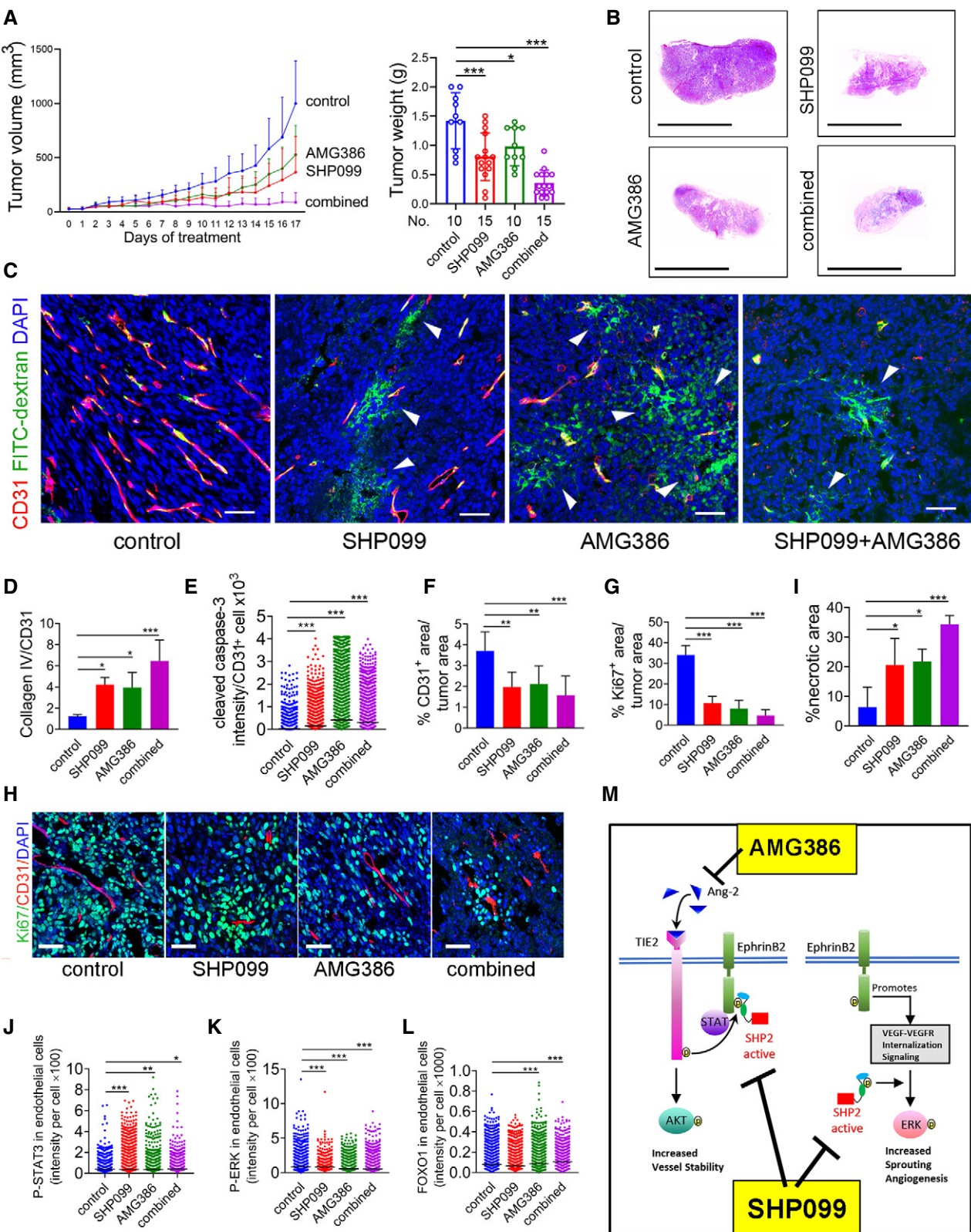

Figure 8.

**Figure 8. SHP099 and AMG386 reduce MC38 tumor growth.**

A, B  AMG386 (5.6 mg/kg s.c.; twice/week) and SHP099 (100 mg/kg orally/daily) were administered individually or together to MC38 tumor-bearing mice (control $n = 10$; SHP099: $n = 15$; AMG386: $n = 10$; SHP099+AMG386 $n = 15$). Tumor growth from initiation of treatment to endpoint (left) and tumor weight (right) at endpoint; error bars ± SD; $P$ values by analysis of variance with Dunnett's multiple comparison test; *$P < 0.05$, **$P < 0.01$, ***$P < 0.001$ (A). Representative control and treated ($n = 4$/group) MC38 tumor sections through the maximum diameter (B); H&E staining; scale bar: 10mm.

C  Vessel perfusion visualized by FITC-dextran (green); CD31 identifies endothelial cells (red); DAPI (blue) identifies the nuclei; representative (of 3 mice/group) confocal images (scale bar: 50 μm); arrow heads point to FITC-dextran extravasation.

D–G  Quantification of "vascular sleeves" (CD31⁻Collagen IV⁺); $n = 4$ tumors/group (D); cleaved caspase-3⁺/CD31⁺ cells; $n = 4$ tumors/group; at least 1000 CD31⁺ cells counted/sample (E); tumor vascular area; $n = 7$ tumors/group (F); Ki67⁺ tumor cells; $n = 4$ tumors/group; at least 1000 CD31⁺ cells counted/sample (G); error bars: ± SD. $P$ values by analysis of variance with Dunnett's multiple comparison test; *$P < 0.05$, **$P < 0.01$, ***$P < 0.001$.

H  Representative images depicting Ki67⁺ cells in control and treated MC38 tumors; representative images (scale bar: 50 μm) of 4 tumors/group.

I  Quantification of necrotic tumor area in control and treated mice; 4 tumors/group; error bars: ± SD. $P$ values by analysis of variance with Dunnett's multiple comparison test; *$P < 0.05$, ***$P < 0.001$.

J–L  Quantification of p-STAT3, p-ERK, and FOXO1 in tumor CD31⁺ endothelial cells in control and treated mice; ($n = 3$ tumors/group); CD31⁺ cells counted in J: control ($n = 3,842$), SHP099 ($n = 21,212$), AMG386 ($n = 12,786$), combined SHP099+AMG386 ($n = 11,746$). CD31⁺ cells counted in K: control ($n = 12,125$), SHP099 ($n = 8,033$), AMG386 ($n = 3,916$), combined SHP099+AMG386 ($n = 4,230$). CD31⁺ cells counted in L: control ($n = 16,076$), SHP099 ($n = 6,555$), AMG386 ($n = 13,715$), combined SHP099+AMG386 ($n = 7,260$). $P$ values by analysis of variance with Dunnett's multiple comparison test; *$P < 0.05$, **$P < 0.01$, ***$P < 0.001$.

M  Schematic representation of the effects of AMG386 and SHP099 on signaling pathways active in the tumor vasculature.

Source data are available online for this figure.

VEGF and anti-Ang1/2 therapies (Scholz *et al*, 2016), perhaps attributable to STAT activation as a consequence of SHP2 inhibition in the tumor vessels (Kreuzaler *et al*, 2011; Salvucci *et al*, 2015). Vigorous approaches to eradication of tumor blood vessels have resulted in tumor necrosis, but also triggered hypoxia-induced vessel regeneration and tumor regrowth (Jain, 2014). Other approaches to targeting the tumor vasculature have focused on "normalizing" the tumor vasculature to reduce tissue hypoxia and improve blood perfusion for proper anti-cancer drug delivery (Jain, 2014). More recently, this approach has been combined with inhibition of neovascularization rather than destruction of existing vasculature (Uribesalgo *et al*, 2019), providing new impetus for the anti-angiogenic treatment of cancer.

The current results show that SHP2 inhibition represents a novel strategy to target endothelial cells in the tumor vasculature and raise the possibility that SHP2 inhibition could be particularly effective when applied more broadly to tumors where SHP2 functions as an oncogenic tyrosine phosphatase.

# Materials and Methods

## Cells, cell culture, and materials

HUVEC (Lifeline Cell Technology, Frederick MD; FC-0003), HDMEC (Clonetics, CC-2543), BMEC (gift from Dr. Jason M. Butler, Weill Cornell Medical College, New York, USA); the murine melanoma B16F10 (ATCC CRL-6475), Lewis lung carcinoma LLC1 (ATCC CRL-1642), mammary cancer 4T1 cell line (ATCC CRL-2539), plasmacytoma MOPC315 (ATCC TIB-23), colon adenocarcinoma MC-38 (a gift of Dr. James Hodge); and human lung carcinoma A549 (ATCC CCL-185), mammary gland carcinoma Hs 578T (ATCC HBT-126), osteosarcoma G-292 (ATCC CRL-1423), NUGC3 gastric adenocarcinoma (JCRB 0822), colon carcinoma RKO (ATCC CRL-2577) and HT29 (ATCC HTB-38), and embryonic kidney 293T (ATCC CRL-3216) were propagated as described in Appendix (Materials and Methods). All cells lines tested *Mycoplasma*-negative but were not authenticated in the laboratory. The inhibitors: SHP099 (Investigational Drugs Repository, DCTD, NCI and MedChemExpress, HY-

100388), AMG386 (Amgen, Inc. under a Materials Cooperative Research and Development Agreement), Tofacitinib (MilliporeSigma, PZ0017), Ruxolitinib (MilliporeSigma, ADV390218177), PD098059 (MedKoo Biosciences, 401680), and SCH772984 (Selleckchem S7101) were used as detailed in Appendix (Materials and Methods). Human recombinant Ang2 (carrier-free) was from BioLegend (753106 or 733104).

## Gene silencing and expression

We generated lentiviral shRNA particles for silencing mouse EphB4 (MISSION shRNA; Sigma-Aldrich TRCN0000023619 and TRCN0000023621); SHP2 (MISSION shRNA; MilliporeSigma, TRCN0000327987, TRCN0000029877, TRCN0000328059, TRCN0000 029875, TRCN0000029878, and TRCN0000327984); controls pLKO (SHC001, no insert) and non-mammalian shRNA (Sigma; SCH002) in 293T cells using the third-generation lentiviral packaging system (Salvucci *et al*, 2015; Kwak *et al*, 2016). Infected cells were selected with puromycin (1 μg/ml, Thermo Fisher, A11138) for 10 days. After RNA purification (RNeasy Micro Kit; QIAGEN 74004), cDNA was synthesized using High-Capacity cDNA Reverse Transcription kit (Applied Biosystems 4368814) and mRNA expression was measured by real-time PCR, as described (1,2,5). Primers for mouse Ang-1 (F: 5'-GACACCTTGAAGGAGGAGAAAG-3'; R: 5'- GCAGGA TGCTGTTGTTGTTG-3'), mouse Ang-2 (F: 5'-GGCTCTCACTGATG GACTTATT- 3'; R:5'-CTGCCACATTCTCTCCTCTCTCTTT-3'), mouse PDGFA (F: 5'- TCCAGCGACTCTTGGAGATA-3'; R:5'-TCTCGGGCAC ATGGTTAATG-3'), mouse FGF2 (F: 5'-CTGCTGGCTTCTAAGTGT GTTA-3'; R: 5'-GCCACATACCAACTGGAGTATT-3') and GAPDH (F: 5'-AGGTCGGTGTGAACGGATTTG-3'; R: 5'- TGTAGACCATGTAGTT GAGGTCA-3') were designed using Primer3. TaqMan probes for mouse EphB1 (Mm00557955_m1; 4331182), EphB2 (Mm01181015_ m1; 4331182), EphB3 (Mm00802553_m1; 4331182), EphB4 (Mm01201157_ml; 4331182), EphB6 (Mm00432456_m1; 4331182), EphA4 (Mm00433056_M1; 4331182), EphrinB2 (Mm00438670_ml; 4331182) and GAPDH (Mm99999915_g1; 4331182) were used with TaqMan Universal PCR Master Mix (Thermo Fisher). Real-time PCR was run on a 7900HT Fast RT PCR System (Applied Biosystems) for 40 cycles with an annealing temperature of 60°C.

## Flow cytometry and cell proliferation

To measure cell viability, cells were incubated (20 min; 37°C) with DRAQ5 (10 μM; Thermo Fisher Scientific, 62251) and DAPI (3 μM; BioLegend, 422801). To measure cell cycle distribution cells were fixed (30 min in cold 70% ethanol at 4°C), washed, treated for 30 min at room temperature with 2.5 μg/ml RNaseA (Thermo Fisher Scientific; 12091039), and stained with propidium iodide (10 μg; Life Technologies, P3566). Flow cytometry analysis was performed on a MoFlo Astrios EQ flow cytometer (Beckman Coulter; B25982) or FACSCantoII (BD Biosciences; 338962). Results were analyzed by FLOJO software (FlowJo LLC). To measure cell proliferation, 0.5 μCi $^3$H- thymidine (NET027WW0011MC; Perkin Elmer) was added to cells in 200 μl of culture medium in 96-well plates for 24 h. Plates were frozen in −80°C, cells collected onto glass fiber filters (Perkin Elmer; 1450-421) and incorporated radioactivity was counted in a liquid scintillation counter (Perkin Elmer; MicroBeta-1450 or MicroBeta-2450).

## ELISAs and cell death proteomic assay

Levels of Ang2 in conditioned medium and in mouse serum were measured by Mouse/Rat Angiopoietin-2 Quantikine ELISA Kit (R&D, MANG20) following the manufacturer's protocol. The colorimetric reaction was measured at 450 nm using a microplate reader (Imgen Technologies). Levels of TNYL-RAW-Fc were measured as described (2) in plasma of mice transduced with TNYL-RAW-Fc plasmid or pcDNA3 vector.

SHP099 modulation of cell death-related proteins in HUVEC was evaluated by a solid-phase antibody array with chemiluminescent detection ("Human Apoptosis Array", R&D Systems, 893900) following the Manufacturer's instructions.

## Immunoprecipitation (IP) and Western blotting

IP and Western blotting were performed essentially as described (Salvucci et al, 2015; Kwak et al, 2016); details and list of antibodies used are in Appendix (Materials and Methods).

## Study approvals and tissue specimens

All animal experiments were approved by the Institutional Animal Care and Use Committee of the CCR, NCI, NIH and conducted in adherence to the NIH Guide for the Care and Use of Laboratory Animals. All human cancer tissue specimens were obtained through approved protocols of the NCI/CCR or through a CCR-approved MTA (Cureline Inc.). Tissue specimens of mouse melanoma from the M1 and M2 models (Marie et al, 2020) were gifts of Drs. A. Sassano, C-P. Day, E. Perez-Guijarro and G. Merlino (CCR/NCI/NIH).

## Animal experiments

Female and male 8-week-old C57BL/6J mice (Jackson Laboratories, 000664), female 10-week-old BALB/cJ mice (Jackson Laboratories, 000651), female 8-week-old NOD-scidIL2Rgamma$^{null}$ mice (Jackson Laboratories, NSG$^{TM}$; 005557) and ), and female 6- to 10-week-old Nu/Nu mice (6- to 10-week-old Charles River Laboratories) were inoculated subcutaneously (s.c.) with B16F10 (C57BL/6J: $1 \times 10^6$

cells; NSG$^{TM}$: $0.5 \times 10^6$), 4T1 ($2 \times 10^6$) or LLC1 ($1.5 \times 10^6$) cells in the abdomen. Tumor volume was estimated by $V = D(d^2)/2$ (D longest and d shortest perpendicular dimensions). Groups of C57BL/6J mice bearing palpable s.c. tumors or NSG$^{TM}$ mice 72 h post-tumor cell inoculation were randomized to receive: daily oral administration (via gavage) of 0.1 ml formulation buffer alone [0.5% Tween 80 (Sigma-Aldrich, P6224) and 0.5% Methyl Cellulose (Sigma-Aldrich, M0430) in distilled water]; daily oral administration of SHP099 (80–200 mg/kg) in 0.1 ml formulation buffer; twice/week s.c. injections of PBS alone (0.1 ml); twice/week s.c. injections of AMG386 (5.6 mg/kg in 0.1 ml PBS); oral daily administration of 0.1 ml formulation buffer + 0.1 ml s.c. PBS twice/week; or daily SHP099 (100–200 mg/kg) in 0.1 ml formulation buffer plus AMG386 (5.6 mg/kg in 0.1 ml PBS) twice/week. Tumor size was measured daily; mouse weight was measured every 2–3 days. All mice were euthanized when tumor(s) in any mouse from any experimental group reached a size of 20 mm in any direction. TNYL-RAW expression in mice was achieved as described (2). To measure blood perfusion, 100 μl of PBS solution containing fluorescein isothiocyanate (FITC)-dextran (2,000,000 MW, 50 mg/ml Thermo Fisher Scientific, D7137) and Poly-L-Lysine (300 kDa, 10 mg/ml Sigma, P-1524) was injected retro-orbitally into mice. After 5 min, the mice were euthanized, and tissues processed. To measure tissue hypoxia, 1.5 mg Hypoxyprobe$^{TM}$- 1 (Hypoxyprobe Inc. HP2-100) was injected retro-orbitally into mice. After 15 min, the mice were euthanized, and tissues processed.

For the B16F10 cell footpad implantation model, $0.5 \times 10^6$ B16F10 cells were inoculated into the footpad region of hind limb of 10-week-old female BALB/cJ mice (Jackson Laboratories, 000651). Four days post-B16F10 cell inoculation, mice were randomized to receive SHP099 (100 mg/kg, orally via gavage, daily) or formulation buffer only, orally via gavage, daily). When the tumor in the hind limb reached a maximum of 10 mm in any direction in any mouse (in either arm), the experiment was terminated. Primary tumor size was estimated from the weight (g) of the hind limb foot resected at the same location from all mice. Popliteal lymph nodes (pLN) ipsilateral to site of injection of B16F10 cells were dissected in toto for further analyses. The lungs were resected for further analyses.

## Immunofluorescence, imaging, and image quantification

Tissue samples were fixed with cold 4% PFA over 72 h at 4°C and processed as described (Salvucci et al, 2015; Kwak et al, 2016). List of antibodies, other reagents, and methods for immunostaining are detailed in Appendix (Materials Methods). For imaging, extended field of view tile image of tissue sections were acquired using a Zeiss LSM780 laser scanning confocal microscope (Carl Zeiss, Oberkochen, Germany) equipped with a 20× plan-apochromat (N.A. 0.8) objective lens and 32-channel GaAsP spectral detector. Details of image acquisition, processing, and analysis are found in Appendix (Materials and Methods).

## Data analysis and statistics

Publicly available datasets are from The Cancer Gene Atlas (TCGA), NCI Genomic Data Commons (GDC) Data Portal (RNA-Seq for colon cancer and HTseq counts for melanoma) and NCBI Gene Expression Omnibus (GSE7553 https://pubmed.ncbi.nlm.nih.gov/18442402/). Informed consent was obtained from all subjects and the

## The paper explained

### Problem
Tumor tissues are dependent upon an adequate vascularization to support progressive expansion and local invasion. Despite considerable efforts at targeting the tumor vasculature to limit the blood supply to cancer tissues, therapeutic benefits have been limited. Increased understanding of endothelial and vascular biology provides new impetus for the anti-angiogenic treatment of cancer.

### Results
Here, we have identified the enzyme SHP2 as an essential factor that supports endothelial cell survival and growth in the remodeling tumor vasculature, but not in the normal resting vasculature. Since endothelial cells form the inner layer of all vessels, extensive damage to endothelial cells impairs vessel function. In melanoma and carcinoma mouse tumor models, we show that systemic treatment with an inhibitor of SHP2, SHP099, promotes degeneration of the tumor vasculature, reduces the number of tumor vessels and tumor oxygenation, and impairs tumor growth. By combining SHP099 with AMG386, which blocks the Angiopoietin/AKT cascade, the vascular and anti-tumor effects of SHP099 are amplified. Since SHP2 and the Angiopoietin/AKT pathways are active in the vascular endothelial cells of human melanoma and colon carcinoma, SHP2 inhibitors alone or with Angiopoietin/AKT inhibitors hold promise as new anti-vascular cancer drugs.

### Impact
The current results show that inhibition of SHP2 represents a new and effective strategy to target endothelial cells in the tumor vasculature of mice without altering the resting normal vasculature. SHP2 inhibition could be particularly effective when applied more broadly to tumors where SHP2 is pro-oncogenic.

experiments conformed to the principles set out in the WMA Declaration of Helsinki and the Department of Health and Human Services Belmont Report. Mann–Whitney *U*-test for comparison of gene expression between tumor and control and Kaplan–Meier survival analysis were performed using statistical computing and graphics software R (version 3.5.2). Results are presented as mean ± SD. No sample was excluded from analysis, and no sample was measured repeatedly. The statistical significance of differences between two groups was calculated using two-tailed Student's *t*-test or Fisher's exact test for analysis of $2 \times 2$ contingency table. Statistical significance among multiple groups was calculated by ANOVA with Dunnett's multiple comparison test. *P* values < 0.05 were considered statistically significant.

Exact *P* values are listed in Appendix Table S1.

## Data availability

This study includes no data deposited in external repositories.

**Expanded View** for this article is available online.

## Acknowledgements

We thank Drs. I. Daar and D. Lowy for generously sharing reagents and helpful discussions; the animal facility personnel, particularly D. Gallardo; L. Lin, K. Stahl, K. Chen, C. Fedele, F. Luo, A. Sassano, C-P. Day, E. Perez-Guijarro, G. Merlino, S. Pittaluga, R. Kines and the personnel of the Laboratory of Cellular Oncology for their help in various aspects of this project. Dr. Wang was a student at West China Medical School, Sichuan University during a period of this research. This work was supported by the intramural program of the Center for Cancer Research, NCI.

## Author contributions

GT conceived and directed the project; YW, OS, and HO designed and executed the experiments; TH, J-XF, HK, YY, and MD helped with the execution of experiments; ADT and MK contributed their expertise to image analyses and image quantification; DW performed bioinformatics analyses; and GT drafted the manuscript; all authors reviewed and contributed to text revisions.

## Conflict of interest

The authors declare that they have no conflict of interest.

## For more information

https://ccr.cancer.gov/Laboratory-of-Cellular-Oncology

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
