## [Review Process File · EMBO Molecular Medicine]

Targeting the SHP2 phosphatase promotes vascular damage and inhibition of tumor growth

Yuyi Wang, Ombretta Salvucci, Hidetaka Ohnuki, Andy Tran, Taekyu Ha, Jing-Xin Feng, Michael DiPrima, Hyeongil Kwak, Dunrui Wang, Yanlin Yu, Michael Kruhlak, and Giovanna Tosato
DOI: [10.15252/emmm.202013543](https://doi.org/10.15252/emmm.202013543)

Corresponding author: Giovanna Tosato (tosatog@mail.nih.gov)

Review Timeline:

Submission Date:	5th Oct 20
Editorial Decision:	30th Oct 20
Revision Received:	4th Feb 21
Editorial Decision:	24th Feb 21
Revision Received:	15th Apr 21
Editorial Decision:	21st Apr 21
Revision Received:	26th Apr 21
Accepted:	4th May 21

Editor: Lise Roth

Transaction Report:

Dear Dr. Tosato,

Thank you for the submission of your manuscript to EMBO Molecular Medicine. We have now heard back from the three referees whom we asked to evaluate your manuscript.

As you will see, the referees all acknowledge the potential interest of the study, however they also have serious and partially overlapping concerns, in particular regarding the need for a second in vivo model, the effects of SHP2 inhibitor on endothelial vs. cancer cells that should be further explored, and the requirement to place the study in the current context of anti-angiogenic therapies and hypoxia-driven pro-tumorigenic effects. As addressing all the referees' comments would require a lot of additional work, time and effort, and as we only invite revisions that are achievable in a reasonable timeframe (~6 months), I am afraid that we cannot offer to consider the manuscript further.

Given the potential interest of the findings, we would, however, be willing to consider a new manuscript on the same topic if at some time in the near future you obtained data that would considerably strengthen the message of the study and address the referees concerns in full.

To be completely clear, however, I would like to stress that if you were to send a new manuscript, this would be treated as a new submission, in particular with respect to the literature and the novelty of your findings at the time of resubmission. If you decide to follow this route, please make sure you nevertheless upload a letter of response to the referees' comments.

I am sorry that I could not bring better news this time, and hope that the referees comments are helpful in your continued work in this area.

Yours sincerely,

Lise Roth

Lise Roth, Ph.D
Editor
EMBO Molecular Medicine

Referee #1 (Comments on Novelty/Model System for Author):

At least two independent cancer models should be studied, rather than B16F10 melanoma mouse model only.

The employed methods do not allow to deconvolve the in vivo SHP2 blockade effects on cancer and endothelial cells.

Referee #1 (Remarks for Author):

In their manuscript, Wang et al. aim at addressing the issue that pharmacological targeting of SHP2 phosphatase in endothelial cells affects angiogenesis along with blood vessel structure and function, thus resulting in reduced tumour perfusion and growth. The authors propose a mechanistic link between the STAT3 pro-apoptotic and the ERK1/2 proliferative pathways and SHP2 inhibition in cultured endothelial cells and in blood vessels of B16F10 syngeneic mouse model of melanoma. They also propose and test in vivo a combination therapy based on the simultaneous inhibition of SHP2 (using SHP099) and Ang2/Tie2 (using AMG386). Albeit the issue of SHP2 inhibition and impact on tumor angiogenesis is potentially interesting, the manuscript suffers of several significant shortcomings.

Major issues

1. In vivo data on the effects of SHP2 and Ang2/Tie2 blockade are obtained from a single B16F10 melanoma mouse model only, when the standard is to investigate in vivo at least two independent cancer models.

2. It is well known that SHP2 inhibitors act not only on endothelial cells, but also on cancer cells. Authors suggest that SHP2 inhibitor SHP099 would differentially act on endothelial and B16F10 melanoma cells. In this regard, the right panel of Fig. 3A is not particularly convincing in showing that SHP099 would decrease pERK1/2, but not increase pSTAT3 in B16F10 melanoma cells. First, when compared to Fig. 3H, pSTAT3 western blot is overexposed in Fig. 3A. Second, above all at 6h, it seems that SHP099 increases pSTAT3 in B16F10 melanoma cells as well. While it is true that in Fig. 2E authors show that SHP099 (1-20 μ M) would minimally affect B16F10 melanoma cell proliferation, based on in vitro pERK1/2/pSTAT3 analyses, it is extremely difficult to deconvolve SHP099 effects on cancer and endothelial cells. Along this line, it seems that in Fig. 4H SHP099 elicits apoptosis (cleaved caspase 3) not only in endothelial cells, but also in surrounding CD31-negative melanoma cells. Authors should show lower magnification images and provide a quantification not only of Ki67 (Fig.4H), but also cleaved caspase-3 in tumor cells. ShRNA-based silencing of SHP2 in BF16F10 cells (Fig. 5) is a more robust and convincing approach that should be used throughout the manuscript. However, to exclude off-targeting effects, the effect of at least two independent SHP2 shRNA should be shown. At present, only the effect of shSHP2 #2 is shown in Fig. 5. A genetic CRISPR/Cas-based KO would be a better approach to deconvolve SHP099 effects on cancer and endothelial cells.

3. Antiangiogenic drugs aimed at blocking tumors' blood supply cause hypoxia, which has been clearly shown to fuel cancer progression and resistance to chemotherapy, radiotherapy, and immunotherapy (reviewed in Jain, *Cancer Cell*, 2014, 26:605-622; Martin et al., *Annu. Rev. Physiol.*, 2019, 81:505-534). This is the reason why much effort is devoted to develop blood vessel targeting drugs that alleviate rather than increase tumor hypoxia (e.g. see Uribealga et al., *EMBO Mol. Med.*, 2019, 11: e9266). Authors show that SHP099 severely disrupts tumor blood vessels and generates large hypoxic areas in BF16F10 tumors. How are authors positioning their study compared to the current literature and vascular modulating therapies in cancer? Can they exclude (as previously reported, e.g. Paez-Ribes et al., *Cancer Cell*, 2009, 15:220-231; Ebos et al., *Cancer Cell*, 2009, 15:232-239; Sennino et al., *Cancer Disc.*, 2012, 2:270-287) that such a severe vascular pruning, while causing primary tumor shrinkage, simultaneously elicit hypoxia-driven metastatic spreading as well (BF16F10 cell injection in the footpad may be used)? Can they evaluate if a standard chemotherapy agent can be effectively delivered both in control and SHP099- or SHP099/AMG386-treated tumors? Are SHP099- or SHP099/AMG386 treatments synergizing with chemotherapy- or immunotherapy-dependent inhibition of primary tumor growth and metastatic dissemination?

Additional issues

1. Western blots shown in Figures 1F and 1H lack of loading control.
2. Why are some data presented as bar plots and other as scatter plots?
3. Figure caption rarely show the number of technical/biological replicates performed to do the statistics. Please add it in.

Referee #2 (Comments on Novelty/Model System for Author):

a second in vivo model is required

Referee #2 (Remarks for Author):

In this manuscripts the authors bring interesting data supporting the concept that SHP2 activated by Ephrin B2/EphB pathway triggers ERK and inhibits STAT-3 resulting in proliferative and anti-apoptotic effect in endothelial cells. These results have been exploited in vivo showing that SHP2 inhibition induced marked effect of vasculature and blood perfusion connected with the reduction of tumor growth in a subset of tumors (eg B16F10 melanoma). Because EphB receptor can be also activated by a trans-phosphorylation mechanism triggered by other TK receptors, including TIE2, the authors demonstrate that the SHP2 inhibition cooperates with the pharmacological block of Ang2/Tie pathway in vascular endothelial cells of human melanoma showing an increased anti-tumor activity. The authors conclude that SHP2 inhibitors can have anti-vascular effects independently from a direct activity on some cancer types of cancer cells, which show a proliferative rate independent from SHP2 inhibition.

The experiments are rigorously planned and conducted, but the major limit of the current version of this MS is that a second mouse model is necessary to better support the authors' conclusions. I appreciated the effort to perform their in vivo experiments in a syngeneic model. In the case that other murine models are not suitable, I suggest to screen human tumor cell lines and find a model with the required features reported in Fig 1. I guess a human model exploited in nude mice should be fine to further demonstrate the vascular, tumor independent effect of SHP2 inhibition.

Figure 1. The confocal pictures are clear but I suggest to quantify the signals for the appropriate statistical analyses. Furthermore in panel F of figure 1, the authors have to show the same analysis performed on tumors induced by B16F10 cells infected with vector alone. Similarly, statistical analysis is required for experiments shown in Supplementary Figure 1 (Panel A). Can the authors exclude that some EphB4-depleted B16F10 cells injected in vivo start to re-express the protein? IF analysis in tumor section might clarify this issue.

Fig 1 H. Literature data demonstrate that Ang2 phosphorylates Tie2 as well as AKT earlier than here reported (e.g. 10.1128/MCB.01472-08; 10.1242/jcs.03077). Is there any explanation? Does sTie2 block the phosphorylating effect of Ang2 on EphrinB2? Does Tie2 silencing inhibit the effect of Ang2 on EphrinB2 phosphorylation? Does Ang1 mimic Ang2 on EphrinB2 phosphorylation?

Fig 2, panel A. Are they serial sections? If it the case it should be declared.

Fig 3. At page 13 the authors write: "In addition, SHP099 activates STAT3 and inhibits ERK activity in endothelial cells beginning at similar early time-points, suggesting that SHP2 may regulate these two pathways directly rather than change in one pathway be compensatory to the other". On my point of view, just the similar time point does not support this statement.

Perhaps that some insights could be achieved by experiments with the concomitant use of MAPK and STAT 3 inhibitors.

Fig 5. By analyzing the data here reported in immune-deficient mice, the authors suggest that SHP099 does not exert an anti-tumor activity. I suggest to analyze macrophage and lymphocyte infiltration in the syngeneic model. This analysis has to be performed too in the mice model treated with the dual therapy (SHP099+ AMG386)

Fig 6, panel E: I suggest to show an illustrative IF image

MINOR POINTS

The following paper (doi.org/10.1161/01.RES.0000199355.32422.7b) should be quoted and discussed because it represents another face of the coin of the role of SHP2 in vascular biology

Legend to Supplementary Fig 2; to Supplementary Fig 3 (panel A and B); to Supplementary Fig 4 (panel G). the number of samples analyzed for each panel has to be indicated

Fig 4: y axis lacks legend

Referee #3 (Remarks for Author):

The paper by Wang et al., focuses on the use of a SHP2 inhibitor to block tumor vascularization and blood perfusion. Blocking of the angiopoietin/tie2/akt cascade increases the effect of SHP2 inhibition.

Although the overall quality of the paper is quite good, the first two parts are not well presented and confusing. The authors attempt drawing a link between EphrinB2 and Ang2. They invest a lot of efforts in finding a model system where activation of EphrinB2 is restricted to the vasculature. Given that the authentic receptors for EphrinB2, namely EphBs, seem not to be involved in this activation, they consider the possibility of a crosstalk between Tie2 and EphrinB2. This hypothesis is based on an educated guess upon examination the literature. Possible cross-talks with other RTKs are unfortunately not investigated.

The activation of EphrinB2 by Ang2 is then directly tested in endothelial cells such as HUVECs. The kinetics of activation is quite surprising since phosphorylation is detected only after 6h of induction with Ang2. In addition, the activation decreases with increasing amounts of Ang2. These results differ from what is usually observed with RTKs like MET or VEGFR2. In these cases, activation can already be detected after a few minutes. These differences in the activation kinetics should have been pointed out. Activation kinetics for other RTKs should be included. There seems to be no difference in the kinetic of activation of Tie2 and EphrinB.

These few in vitro experiments are not enough to show a strong link between angiopoietin and EphrinB2. A crosstalk with other RTKs including VEGFR2 or MET should have been examined. Co-immunoprecipitation experiments or proximity ligation assays should be performed in order to examine the possible link between EphrinB2 and Tie2 more in depth. Kinase dead mutant of Tie2 might also be included in the study.

The authors proposed the B16F10 melanoma as the best system for their study but switched in this second part of the paper to other types of cancers. In principle it is a good thing but

The examination of the correlation of VEGFA and Ang 2 expression with survival is not well justified. This part is very confusing.

The in vitro experiments do show an impact of the SHP2 inhibitor on endothelial cell proliferation and cell death. The link to STAT3 activation and ERK inhibition is well illustrated. However, the quality of the blots in Figure 3 should be improved. There are some inconsistencies between the 2 endothelial cell lines in the SHP2 inhibitor treatment (for example after 24h in Fig3A). In the case of Trametinib, the basal level of p-ERK is not always consistent.

The in vivo experiments using the inhibitors are convincing and indeed show a decreased tumor vascularization upon treatment with both reagents (SHP099 and AMG386).

Altogether the combination of these two inhibitors might be useful since when used together they convincingly show improved results. A comparison with the use of bevacizumab would be interesting.

Referee #1 (Comments on Novelty/Model System for Author):

At least two independent cancer models should be studied, rather than B16F10 melanoma mouse model only.

As detailed below, we have now studied an additional cancer model (MC38 mouse colorectal cancer) and report these results in the revised manuscript.

The employed methods do not allow to deconvolve the in vivo SHP2 blockade effects on cancer and endothelial cells.

As suggested by the Reviewer, we now provide additional evidence to distinguish the effects of SHP2 blockade in tumor and endothelial cells.

Referee #1 (Remarks for Author):

In their manuscript, Wang et al. aim at addressing the issue that pharmacological targeting of SHP2 phosphatase in endothelial cells affects angiogenesis along with blood vessel structure and function, thus resulting in reduced tumour perfusion and growth. The authors propose a mechanistic link between the STAT3 pro-apoptotic and the ERK1/2 proliferative pathways and SHP2 inhibition in cultured endothelial cells and in blood vessels of B16F10 syngeneic mouse model of melanoma. They also propose and test in vivo a combination therapy based on the simultaneous inhibition of SHP2 (using SHP099) and Ang2/Tie2 (using AMG386). Albeit the issue of SHP2 inhibition and impact on tumor angiogenesis is potentially interesting, the manuscript suffers of several significant shortcomings.

We thank the Reviewer for the time and effort to review our manuscript. We appreciate that the Reviewer found the impact of SHP2 inhibition on tumor angiogenesis as “potentially interesting” and pointed to several shortcomings. We have addressed each of the constructive comments of the Reviewer and now provide a revised manuscript that addresses all the comments.

Major issues

1. In vivo data on the effects of SHP2 and Ang2/Tie2 blockade are obtained from a single B16F10 melanoma mouse model only, when the standard is to investigate in vivo at least two independent cancer models.

We now provide new results on the effects of SHP2 and Ang2/Tie2 blockade in the additional MC38 syngeneic colon cancer model. The new results confirm the anti-vascular and anti-cancer activities of SHP2 and Ang2/Tie2 blockade described in the B16F10 melanoma model.

Briefly, the mouse MC38 colon carcinoma cell line is similar to B16F10 in being SHP099 growth resistant (Supplementary Fig. 4G) and generating Ang2⁺ tumors in syngeneic mice showing EphrinB, Tie2 and SHP2 phosphorylation in the tumor vasculature (new Supplementary Figure 9A). SHP099 and AMG386, individually and cooperatively reduced the growth and increased cell death in MC38 tumors (new Fig. 8A, B, G, H and I). In addition, SHP099 and AMG386 display prominent anti-vascular effects in MC38 tumors as evidenced by increased vessel permeability, endothelial cell death and vascular degeneration (new Fig. 8 C-F,

and new Supplementary Fig. 9B). These anti-vascular effects are associated with a significant increase of vascular p-STAT3 and reduction of p-ERK (Fig. 8 J,H; Supplementary Fig. 9F). AMG386 reduces vascular p-Tie2 and p-EphrinB (Supplementary Fig. 9 C-E), and increases vascular FOXO1 (Fig. 8L, Supplementary Fig. 9F).

2. It is well known that SHP2 inhibitors act not only on endothelial cells, but also on cancer cells. Authors suggest that SHP2 inhibitor SHP099 would differentially act on endothelial and B16F10 melanoma cells. In this regard, the right panel of Fig. 3A is not particularly convincing in showing that SHP099 would decrease pERK1/2, but not increase pSTAT3 in B16F10 melanoma cells. First, when compared to Fig. 3H, pSTAT3 western blot is overexposed in Fig. 3A. Second, above all at 6h, it seems that SHP099 increases pSTAT3 in B16F10 melanoma cells as well. While it is true that in Fig. 2E authors show that SHP099 (1-20 μ M) would minimally affect B16F10 melanoma cell proliferation, based on in vitro pERK1/2/pSTAT3 analyses, it is extremely difficult to deconvolve SHP099 effects on cancer and endothelial cells. Along this line, it seems that in Fig. 4H SHP099 elicits apoptosis (cleaved caspase 3) not only in endothelial cells, but also in surrounding CD31-negative melanoma cells. Authors should show lower magnification images and provide a quantification not only of Ki67 (Fig.4H), but also cleaved caspase-3 in tumor cells. ShRNA-based silencing of SHP2 in B16F10 cells (Fig. 5) is a more robust and convincing approach that should be used throughout the manuscript. However, to exclude off-targeting effects, the effect of at least two independent SHP2 shRNA should be shown. At present, only the effect of shSHP2 #2 is shown in Fig. 5. A genetic CRISPR/Cas-based KO would be a better approach to deconvolve SHP099 effects on cancer and endothelial cells.

The Reviewer recommended that we strengthen the evidence for B16F10 resistance to SHP2 blockade and suggested appropriate experiments. We have followed all the suggestions, as described below.

a. We have validated 5 shRNAs as effective at depleting SHP2 in endothelial cells (BMEC) and in B16F10 cells (new Fig. 2F). We now show that the effective depletion of SHP2 with 3 of these shRNAs reduces substantially endothelial cell proliferation but has no inhibitory effect on the proliferation of B16F10 cells (new Fig. 2G). We further confirmed that B16F10 melanoma cells are SHP2-growth independent by maintaining a long-term B16F10 melanoma line in which SHP2 is persistently undetected (described in the text). These new results support the conclusion that endothelial cells are dependent on SHP2 for growth whereas the B16F10 melanoma cells are growth independent from SHP2 and serve to complement the experiments in mice (Fig. 5D-G). In vivo, SHP2-depleted B16F10 tumors (persistent depletion of SHP2 was confirmed at the end of the experiment by analyzing the tumors, Fig. 5G) and control B16F10 tumors (not depleted of SHP2) are similarly responsive to SHP099 (Fig. 5D and E), supporting a role of SHP099 in the tumor microenvironment rather than on the tumor cells.

b. We have substituted the image in Figure 3A from B16F10 immunoblotting with an improved image.

c. The Referee correctly noted in Fig. 4H the presence of tumor cell death (cleaved caspase 3⁺ cells). As suggested, we have now added lower magnification images of control and SHP099 treated tumors (Supplementary Fig. 5E) and provide quantitative results of cell death in control and SHP099-treated tumors (Supplementary Fig. 5F). Increased tumor cell death after treatment

with SHP099 is likely vascular-based, consistent with the extensive vascular damage (Fig. 4D, E, H and I) and tumor ischemia (Fig. 4L) caused by SHP099.

3. Antiangiogenic drugs aimed at blocking tumors' blood supply cause hypoxia, which has been clearly shown to fuel cancer progression and resistance to chemotherapy, radiotherapy, and immunotherapy (reviewed in Jain, Cancer Cell, 2014, 26:605-622; Martin et al., Annu. Rev. Physiol., 2019, 81:505-534). This is the reason why much effort is devoted to develop blood vessel targeting drugs that alleviate rather than increase tumor hypoxia (e.g. see Uribealgo et al., EMBO Mol. Med., 2019, 11: e9266). Authors show that SHP099 severely disrupts tumor blood vessels and generates large hypoxic areas in BF16F10 tumors. How are authors positioning their study compared to the current literature and vascular modulating therapies in cancer? Can they exclude (as previously reported, e.g. Paez-Ribes et al., Cancer Cell, 2009, 15:220-231; Ebos et al., Cancer Cell, 2009, 15:232-239; Sennino et al., Cancer Disc., 2012, 2:270-287) that such a severe vascular pruning, while causing primary tumor shrinkage, simultaneously elicit hypoxia-driven metastatic spreading as well (BF16F10 cell injection in the footpad may be used)? Can they evaluate if a standard chemotherapy agent can be effectively delivered both in control and SHP099- or SHP099/AMG386-treated tumors? Are SHP099- or SHP099/AMG386 treatments synergizing with chemotherapy- or immunotherapy-dependent inhibition of primary tumor growth and metastatic dissemination?

The Reviewer raises an important topic in the field of anti-vascular approaches to cancer, i.e. to which extent vigorous approaches at eradicating tumor blood vessels that result in tumor shrinkage, can then trigger hypoxia-driven resistance mechanisms that drive tumor regrowth. A contrasting approach is to promote vessel normalization in tumors, reducing increased permeability and improving vessel structure, without tumor hypoxia. VEGF/VEGFR targeting regimens have produced short-lived vessel normalization. More recently, the Penninger's group has identified Apelin as a potential drug target that reduces tumor vascularization while "normalizing" tumor vessels in selected tumor types, without resulting in increased tumor hypoxia. This exciting research provides welcome new incentive for the anti-angiogenic treatment of cancer.

To our knowledge, a promising anti-vascular approach currently being tested in phase III trials is the combination of VEGF and Ang2 blockade, which increases tumor hypoxia (Rigamonti et al Cell Rep 8:696, 2014 and Scholz et al, EMBO Mol Med 8:39, 2016; Schmittnaegel M et al Sci Transl Med 9: eaak9660, 2017), but has nonetheless shown to effectively decrease tumor vascularization, tumor growth, and metastasis in pre-clinical models. In addition, a phase III trial with AMG386 (Trebuanib) in conjunction with Paclitaxel and Carboplatin chemotherapy (TRINOVA-3: A Study of AMG 386 in Combination with Paclitaxel and Carboplatin to Treat Ovarian Cancer; Vergote, I. et al; the Lancet Oncology 20: 862, 2019) has shown that AMG386 does not negatively impact clinical benefit from Paclitaxel and Carboplatin chemotherapy, despite inducing tumor hypoxia. We thank the Reviewer for highlighting this central theme in the antiangiogenic treatment of cancer and prompting a discussion in light of our new approach that combines Ang2 and SHP2 targeting (see Discussion, page 25).

We are very interested at combining SHP099 +/- AMG386 with chemotherapy in pre-clinical models, as suggested by the Reviewer. In the current study, we have selected tumors insensitive to SHP099 with the goal to highlight the anti-vascular role of SHP099. Since most mouse tumors evaluated for the current studies show that both the tumor endothelium and the

tumor cells are sensitive to SHP099 inhibition, in the next phase of our studies we want to examine the combined effects of SHP099 on the tumor cells and the tumor endothelium. In this more complex, but more realistic context, we would like to combine chemotherapy. Respectfully, we would like to defer incorporating chemotherapy into our next series of experiments.

Additional Issues

1. Western blots shown in Figures 1F and 1H lack of loading control.

In the interest of space, we have shown EphrinB2 as a loading control, as it also serves as a control for p-EphrinB; we provide the Reviewer with the additional results of β -actin loading control (for each of the 7 membranes; one per time-point, which confirms equal loading (attachment for Reviewer no. 1)

2. Why are some data presented as bar plots and other as scatter plots?

We have used scatter plots when the results denote fluorescence intensity/cell; we have used bar graphs when the results denote positive areas/total areas.

3. Figure caption rarely show the number of technical/biological replicates performed to do the statistics. Please add it in.

We have done so.

Referee #2 (Comments on Novelty/Model System for Author):

a second in vivo model is required

Referee #2 (Remarks for Author):

In this manuscript the authors bring interesting data supporting the concept that SHP2 activated by Ephrin B2/EphB pathway triggers ERK and inhibits STAT-3 resulting in proliferative and anti-apoptotic effect in endothelial cells. These results have been exploited in vivo showing that SHP2 inhibition induced marked effect of vasculature and blood perfusion connected with the reduction of tumor growth in a subset of tumors (eg B16F10 melanoma). Because EphB receptor can be also activated by a trans-phosphorylation mechanism triggered by other TK receptors, including TIE2, the authors demonstrate that the SHP2 inhibition cooperates with the pharmacological block of Ang2/Tie pathway in vascular endothelial cells of human melanoma showing an increased anti-tumor activity. The authors conclude that SHP2 inhibitors can have anti-vascular effects independently from a direct activity on some cancer types of cancer cells, which show a proliferative rate independent from SHP2 inhibition. The experiments are rigorously planned and conducted, but the major limit of the current version of this MS is that a second mouse model is necessary to better support the authors' conclusions. I appreciated the effort to perform their in vivo experiments in a syngeneic model. In the case that other murine model are not suitable, I suggest to screen human tumor cell lines and find a model with the required features reported in Fig 1. I guess a human model exploited in nude mice should be fine to further demonstrate the vascular, tumor independent effect of SHP2 inhibition.

We thank the Reviewer for the time and effort in providing constructive suggestions for improvement, and for considering our experiments “rigorously planned and conducted”.

The Reviewer asked that we support the observations made in B16F10 syngeneic mouse model with a second mouse model. We have identified the murine MC38 colon carcinoma cell line as being growth resistant to SHP2 inhibition with SHP099 comparably to B16F10 murine melanoma (Suppl. Fig. 4G). We have used the MC38 syngeneic mouse tumor model to compare the results with the B16F10 syngeneic mouse tumor model. The new in vivo results (Figure 8 and Supplementary Fig. 9) confirm the anti-vascular and anti-tumor activity of SHP099 and the enhanced activity of treatment with SHP099 plus AMG386. They further confirm the biochemical consequences of these treatments.

Figure 1. The confocal pictures are clear but I suggest to quantify the signals for the appropriate statistical analyses.

We have now quantified the fluorescence signals shown in Figure 1 (all panels) and provide the results in the text (p-EphrinB/CD31 (panel A, B and G); p-EphrinB/NG2-positive pericytes (panel C); CD31/pTIE2 (panel I) and CD31/p-SHP2 (panel I).

Furthermore in panel F of figure 1, the authors have to show the same analysis performed on tumors induced by B16F10 cells infected with vector alone.

The results requested by the Reviewer are shown in Fig. 1F. These include analysis of tumors induced by B16F10 cells infected with vector alone (pLKO; n=2) and with the EphB4-specific shRNA (shEphB4; n=2). These results are representative of 8 mice/group.

Similarly, statistical analysis is required for experiments shown in Supplementary Figure 1 (Pane A).

We now provide the analysis for p-EphrinB/CD31 and p-SHP2/CD31 shown in Supplementary Fig. 1A and provide all the results in the text.

Can the authors exclude that some EphB4-depleted B16F10 cells injected in vivo start to re-express the protein? IF analysis in tumor section might clarify this issue.

We have verified by immunoblotting that each of 10 tumors from Eph4-depleted B16F10 cells continued to be EphB4 depleted once removed from the mice (including the tumors used for immunohistochemistry in Fig. 1G and I), whereas each of 10 tumors from vector control pLKO-infected B16F10 expressed EphB4. An example (2 tumors/group) is shown in the immunoblotting results in Fig. 1F. Thus, the tumor vasculature is p-EphrinB-positive in the continued absence of tumor-associated EphB4 (depleted by shRNA).

Fig 1 H. Literature data demonstrate that Ang2 phosphorylates Tie2 as well as AKT earlier than here reported (e.g. 10.1128/MCB.01472-08; 10.1242/jcs.03077). Is there any explanation? Does sTie2 block the phosphorylating effect of Ang2 on EphrinB2? Does Tie2 silencing inhibit the effect of Ang2 on EphrinB2 phosphorylation? Does Ang1 mimic Ang2 on EphrinB2 phosphorylation?

We agree with the Reviewer that our kinetics of Ang2-induced Tie2 activation in endothelial cells (HUVEC) are delayed when compared to the kinetics of activation of other receptor tyrosine kinases. We also recognize that other investigators reported a more rapid Ang2-induced activation of Tie2. For example, Bogdanovic E et al (J. Cell Sci 119: 3552, 2006; using Ang2 containing a polyhistidine tag from R&D Systems) and al Yuan H.T. et al (Mol Cell Biol 29: 2011, 2009; from R&D Systems; no other information) reported Tie2 phosphorylation 15 min and 30 min after HUVEC incubation with Ang2.

However, our results using human recombinant Ang2 from BioLegend (cat no. 753106 or 753104) are consistent with other published results. Teichert-Kuliszewska et al (Cardiovascular Res 49: 659, 2000) showed that Ang2 does not induce Tie2 phosphorylation in HUVEC after 5 minutes or after 1-hour incubation; only after 24 hr did Ang2 induce vigorous Tie2 phosphorylation in HUVEC, particularly when more Ang2 was added for 5 minutes after 24 hr pre-incubation.

It seems that Ang2 multimerization status and other physicochemical characteristics may be critical for Tie2 activation. As much as 80-90% of recombinant Ang2 consisted of trimeric, tetrameric or pentameric oligomers in an Ang2 preparation that induced Tie2 phosphorylation after 15 min incubation (Kim K. et al JBC 280: 20126, 2005). Instead, recombinant Ang2 preparations that consisted of disulfide-linked dimers did not activate Tie2 (Yancopoulos G. D. et al Nature 407: 242, 200; Procopio W.N. et al JBC 274: 30196, 1999). We have enquired as to the oligomerization status of the Ang2 preparation from BioLegend we used; technical service wrote: "it is composed of monomers, dimers and others", but % monomers was not provided.

The literature has overall described Ang2 as a highly "context-dependent" antagonist or agonist of Tie2, but the reasons for these contextual differences are still under investigation

(Thurston G and Daly C. Cold Spring Harb Perspect Med 2:a006650, 2012; Saharinen P. et al Nat Rev Drug Discov 16:635, 2017). Initially, Ang2 was considered a pure antagonist of Ang1, the natural agonist of Tie2, competitively inhibiting Ang1 binding to Tie2; Ang2 was reported unable to activate Tie2 in multiple studies (Maisonpierre P.C et al Science 277: 55, 1997; Procopio WN et al J B C 274: 30196,1999; Saharinen P. et al JCB 169: 239, 2005). Subsequent discoveries have unequivocally shown that Ang2 can function as an agonist of Tie2 (Kim K. et al JBC 280: 20126, 2005; Thurston and Daly, 2012; Yuan et al, 2019). Many key factors are likely involved in Ang2/Tie2 activation, including the interplay among the four Tie2 ligands (Ang1-4), oligomerization status of Ang2 (dimers, trimers and tetramers), receptor availability (particularly Tie1 regulation of Tie2), Tie2 localization (particularly at cell-to-cell junctions), internalization and degradation (particularly the contribution of the vascular endothelial protein tyrosine phosphatase, VE-PTP) and receptor ectodomain cleavage (this topic was recently reviewed in Zhang Y. et al iScience 20: 497-511, October 2019).

In response to the specific questions of the Reviewer, BioLegend includes sTie2 blocking of Ang2 biological activity as part of release assays for Ang2 lots; we felt it unnecessary to repeat.

Regarding Ang1 inducing EphrinB2 activation, original observations showed that active Tie2 (regardless the source of the activity, constitutive or ligand-induced) can phosphorylate the intracellular domain of EphrinB2 (Adams R. H. Genes and Dev 13:295, 1999). Based on this information, the fact that Ang1 is a strong activator of Tie2 and our observation that Ang1 is not expressed in B16F10 (Supplemental Fig. 2C), we have elected not to test Ang1.

Fig 2 , panel A. Are they serial sections? If it the case it should be declared.

Yes, serial sections were used for the 4 stains: CD31/p-EphrinB; CD31/p-SHP2; CD31/p-Tie2 and CD31/Ang2; this is now stated in the legend of Fig. 2A.

Fig 3. At page 13 the authors write: "In addition, SHP099 activates STAT3 and inhibits ERK activity in endothelial cells beginning at similar early time-points, suggesting that SHP2 may regulate these two pathways directly rather than change in one pathway be compensatory to the other". On my point of view, just the similar time point does not support this statement. Perhaps that some insights could be achieved by experiments with the concomitant use of MAPK and STAT 3 inhibitors.

We agree with the Reviewer that the similar kinetics of SHP099 activation of p-STAT3 and inhibition of p-ERK1/2 in endothelial cells is consistent with SHP2 directly activating these two pathways, rather than providing evidence. We have changed the text to read: "...raising the possibility that SHP2 may directly regulate these two pathways rather than change in one pathway be compensatory to the other."

Also, as suggested by the Reviewer, we have now used STAT and MAPK inhibitors to gain additional insights. These new results are shown in the Figure attached (labeled Figure for Reviewer no. 2, panels A-D). In sum, the results show that chemical inhibition of STAT signaling not only inhibits STAT signaling but also activates ERK1/2 in endothelial cells, and that chemical inhibition of ERK1/2 signaling not only reduces pERK1/2 but also activates STAT3 in endothelial cells. This compensatory modulation of alternative signaling pathways has been noted previously (Sakaguchi M et al J Invest Dermatol 2012; 132:1877-1885; Vultur A et al

Oncogene 2014; 33:1850-61) and prevents gaining meaningful mechanistic insights on how SHP099 modulates MAPK and STAT3 signaling. For this reason, we have elected not to include these results in the paper.

Fig 5. By analyzing the data here reported in immune-deficient mice , the authors suggest that SHP099 does not exert an anti-tumor activity. I suggest to analyze macrophage and lymphocyte infiltration in the syngeneic model. This analysis has to be performed too in the mice model treated with the dual therapy (SHP099+ AMG386)

We have now examined macrophage (F4/80) and lymphocyte (CD3-T cells and CD19-B cells) infiltration in B16F10 and MC38 tumors established in immunocompetent syngeneic B6 mice. F4/80⁺ macrophage infiltrates are detected in both tumors types and they change insignificantly with treatment (Supplementary Fig 5G and Supplementary Fig 9G). CD3⁺ T cell infiltrates, rarely detected in B16F10 tumors, are clearly observed in MC38 tumors and they change insignificantly with treatment (Supplementary Fig. 9H). CD19⁺ cells are virtually undetected in both B16F10 and MC38 tumor tissues.

Fig 6, panel E: I suggest to show an illustrative IF image

We now provide an illustrative IF image from these results (new Figure 6E).

MINOR POINTS

The following paper (doi.org/10.1161/01.RES.0000199355.32422.7b) should be quoted and discussed because it represents another face of the coin of the role of SHP2 in vascular biology

We thank the Reviewer for pointing to this very interesting paper by Mitola S et al (Circ Res. 98: 45, 2006), which highlights the role of SHP2 as an essential of vitronectin-induced inhibitor of VEGFA-VEGFR2 activation. We now quote (ref 52) and discuss this very relevant paper.

Legend to Supplementary Fig 2; to Supplementary Fig 3 (panel A and B) ; to Supplementary Fig 4 (panel G) .the number of samples analyzed for each panel has to be indicated

We now provide sample numbers to the legends of Supplemental Figures 2-4 where missing.

Fig 4: y axis lacks legend

Corrected the error.

Referee #3 (Remarks for Author):

The paper by Wang et al., focuses on the use of a SHP2 inhibitor to block tumor vascularization and blood perfusion. Blocking of the angiopoietin/tie2/akt cascade increases the effect of SHP2 inhibition.

Although the overall quality of the paper is quite good, the first two parts are not well presented and confusing. The authors attempt drawing a link between EphrinB2 and Ang2. They invest a lot of efforts in finding a model system where activation of EphrinB2 is restricted to the vasculature. Given that the authentic receptors for EphrinB2, namely EphBs, seem not to be involved in this activation, they consider the possibility of a crosstalk between Tie2 and EphrinB2. This hypothesis is based on an educated guess upon examination the literature.

Possible cross-talks with other RTKs are unfortunately not investigated.

The activation of EphrinB2 by Ang2 is then directly tested in endothelial cells such as HUVECs. The kinetics of activation is quite surprising since phosphorylation is detected only after 6h of induction with Ang2. In addition, the activation decreases with increasing amounts of Ang2.

These results differ from what is usually observed with RTKs like MET or VEGFR2. In these cases, activation can already be detected after a few minutes. These differences in the activation kinetics should have been pointed out. Activation kinetics for other RTKs should be included.

There seems to be no difference in the kinetic of activation of Tie2 and EphrinB.

These few in vitro experiments are not enough to show a strong link between angiopoietin and EphrinB2. A crosstalk with other RTKs including VEGFR2 or MET should have been examined. Co-immunoprecipitation experiments or proximity ligation assays should be performed in order to examine the possible link between EphrinB2 and Tie2 more in depth. Kinase dead mutant of Tie2 might also be included in the study.

The authors proposed the B16F10 melanoma as the best system for their study but switched in this second part of the paper to other types of cancers. In principle it is a good thing but

The examination of the correlation of VEGFA and Ang 2 expression with survival is not well justified. This part is very confusing.

The in vitro experiments do show an impact of the SHP2 inhibitor on endothelial cell proliferation and cell death. The link to STAT3 activation and ERK inhibition is well illustrated. However, the quality of the blots in Figure 3 should be improved. There are some inconsistencies between the 2 endothelial cell lines in the SHP2 inhibitor treatment (for example after 24h in Fig3A). In the case of Trametinib, the basal level of p-ERK is not always consistent.

The in vivo experiments using the inhibitors are convincing and indeed show a decreased tumor vascularization upon treatment with both reagents (SHP099 and AMG386).

Altogether the combination of these two inhibitors might be useful since when used together they convincingly show improved results. A comparison with the use of bevacizumab would be interesting.

We thank the Reviewer for reviewing the manuscript and providing insightful comments. We are pleased that the Reviewer considers the overall quality of our paper as “quite good”.

Below, we have extracted and addressed each point made by the Reviewer

1. *The kinetics of activation is quite surprising since phosphorylation is detected only after 6h of induction with Ang2. In addition, the activation decreases with increasing amounts*

of Ang2. These results differ from what is usually observed with RTKs like MET or VEGFR2. In these cases, activation can already be detected after a few minutes. These differences in the activation kinetics should have been pointed out. Activation kinetics for other RTKs should be included. There seems to be no difference in the kinetic of activation of Tie2 and EphrinB.

We agree with the Reviewer that the kinetics of Ang2 activation of Tie2 are surprisingly delayed when compared to the kinetics of activation of other RTK. However, they are consistent with other published results. For example, Ang2 did not induce Tie2 phosphorylation in HUVEC after 5 minutes or after 1-hour incubation; after 24 hr, Ang2 induced vigorous Tie2 phosphorylation, particularly when additional Ang2 was added for 5 minutes (Teichert-Kuliszewska K. et al. Cardiovascular Res 49: 659, 2000). Other studies have reported that when present in trimeric, tetrameric and pentameric forms, recombinant Ang2 induces Tie2 phosphorylation in serum starved HUVEC after 15 minutes incubation, but not as potently as Ang1 (Kim K. et al JBC 280: 20126, 2005). Pre-clustered Ang2 (400-1000 ng/ml) also induced Tie2 phosphorylation in endothelial cells after 15 min, although Tie2 phosphorylation was lower than induced by Ang1 (Bogdanovic E et al J Cell Sci 119: 3551, 2006).

The literature has overall described Ang2 as a highly “context-dependent” antagonist or agonist of Tie2 (Thurston G and Daly C. Cold Spring Harb Perspect Med 2:a006650, 2012; Saharinen P. et al Nat Rev Drug Discov 16:635, 2017). Initially, Ang2 was considered a pure antagonist of Ang1, the natural agonist of Tie2, competitively inhibiting Ang1 binding to Tie2; Ang2 was reported unable to activate Tie2 in a number of studies (Maisonpierre P.C et al Science 777: 55, 1997; Saharinen P. et al JCB 169: 239, 2005). Subsequent discoveries have unequivocally shown that Ang2 can function as an agonist of Tie2 (Kim K. et al JBC 280: 20126, 2005; Thurston and Daly, 2012; Yuan et al, 2019).

Tie2 activation by Ang2 appears complex. It is clear that that many key factors are involved, including the interplay among the four Tie2 ligands (Ang1-4), oligomerization state of Ang2 (dimers, trimers and tetramers), receptor availability (particularly Tie1 regulation of Tie2), Tie2 localization (particularly at cell-to-cell junctions), internalization and degradation (particularly the contribution of the vascular endothelial protein tyrosine phosphatase, VE-PTP) and receptor ectodomain cleavage (a review of this topic is found in Zhang Y. et al iScience 20: 497-511, October 2019).

In sum, the kinetics of Ang2 activation of Tie2 in endothelial cells in our current study are consistent with other results in the literature, and clearly differ from the usually observed activation kinetics of other RTKs in endothelial cells. As suggested by the Reviewer, we now point in the text to the kinetic differences between Ang2 activation of Tie2, and the rapid activation of other RTKs by their ligands. We also provide a typical example of early RTK activation (VEGFA activates VEGFR2 phosphorylation (Tyr1175) in HUVEC within minutes; Supplementary Fig. 2G).

- 2. There seems to be no difference in the kinetic of activation of Tie2 and EphrinB.* Indeed, the timing of Tie2 and EphrinB2 phosphorylation is similar, consistent with the occurrence of rapid crosstalk between the Ang2-activated Tie2 RTK and EphrinB2 phosphorylation. Similar to these results, Bruckner et al. (Science 275: 1640, 1997)

showed that PDGF activation of PDGFR RTK and EphrinB2 (then called Lerk) phosphorylation were both observed after 1 min with PDGF, with no delay between PDGFR and EphrinB2 phosphorylation. The rapid kinetics of EphrinB2 and Tie2 or PDGFR phosphorylation suggested that EphrinB2 is a direct target of the Tie2 and PDGF RTK.

3. *Possible cross-talks with other RTKs are unfortunately not investigated. A crosstalk with other RTKs including VEGFR2 or MET should have been examined. Co-immunoprecipitation experiments or proximity ligation assays should be performed in order to examine the possible link between EphrinB2 and Tie2 more in depth. Kinase dead mutant of Tie2 might also be included in the study.*

As indicated by the Reviewer crosstalk of EphrinB2 with other RTKs is an important aspect. In the submission, we did not to present the results of our broad and unbiased consideration of all RTKs, and focused instead on Tie2, which had emerged as a leading candidate from our broader analysis. Briefly, our approach included the following steps:

- i. We used a network algorithm (GPS 3.0) that uses kinase information to predict the probability of phosphorylation of a target peptide. The target peptide in EphrinB2 consists of residues 251-322 in the intracellular domain of EphrinB2. The results of this analysis identified 48 unique receptor kinases/kinases with the potential to phosphorylate the intracellular domain of EphrinB2 (residues 251-322).
- ii. Based on publicly available gene expression results from murine dermal microvascular endothelial cells (GSM 1439480), we evaluated endothelial expression of the 48 candidate receptor kinases/kinases. Based on positive and negative controls for gene expression in endothelial cells, 21 genes were expressed well, 11 were expressed at low levels and 16 were essentially not expressed. TEK/TIE2, FLT1/VEGFR1, KDR/VEGFR2 and FLT4/VEGFR3 were among the 21 receptor kinases/kinases expressed well in endothelial cells.
- iii. To prioritize among these kinases/receptor kinases, we examined differential gene expression in three murine cancer cell lines (B16F10, LLC1 and 4T1), only one of which (B16F10) supported EphrinB2 phosphorylation in the tumor vasculature. We identified 481 genes differentially expressed in B16F10. Only 32 of these coded for soluble factors or transmembrane proteins, the likely ligands of receptor kinases in endothelial cells. Among these, B16F10 expressed Ang2 at significantly higher levels than LLC1 and 4T1. In addition, mice bearing B16F10 tumor had significantly higher levels of Ang2 in the circulation.
- iv. Based on this convergence, we hypothesized that secreted Ang2 can activate Tie2 in our tumor model, and that Tie2 can trans-phosphorylate EphrinB2.

We now provide to the Reviewer (attachment for Reviewer 3) a summary of the results of our broad analysis of endothelial RTKs/kinases potentially involved in crosstalk with EphrinB2 (attached here). We have outlined these experiments in the text (page 9) and provide some of these results in Supplementary Figure 2A,B.

4. *The examination of the correlation of VEGFA and Ang 2 expression with survival is not well justified. This part is very confusing.*

We have now explained in greater detail why we examined the correlation between expression levels of Ang2 and VEGFA with patient survival from melanoma and colon cancer. In brief, we

had identified a p-EphrinB2 dependent pathway for endothelial cell survival and found that p-EphrinB2 is variably active in the vascular endothelial cells of melanoma and colon carcinoma but to a lower degree in other human cancer types. Hence, we examined whether the degree of activity of this pathway in human melanoma and colon carcinoma might correlate with tumor growth. Since cancer patient survival is a surrogate for tumor growth, we looked to correlate cancer patient survival with gene expression of potential activators of EphrinB2. We focused on ANGPT1, ANGPT2, VEGFA, PDGFA and FGF2 because these are ligands of RTKs that can cross-phosphorylate EphrinB2.

- 5. However, the quality of the blots in Figure 3 should be improved. There are some inconsistencies between the 2 endothelial cell lines in the SHP2 inhibitor treatment (for example after 24h in Fig3A). In the case of Trametinib, the basal level of p-ERK is not always consistent.*

We now provide images of new and improved blots in Figure 3A. We recognize that there are some inconsistencies in the basal levels of p-ERK (which we believe are attributable to fluctuations of p-ERK under basal conditions). To mitigate the impact of these inconsistencies for evaluation of the results, we include the basal p-ERK levels at each experimental time-point.

- 6. The in vivo experiments using the inhibitors are convincing and indeed show a decreased tumor vascularization upon treatment with both reagents (SHP099 and AMG386). Altogether the combination of these two inhibitors might be useful since when used together they convincingly show improved results. A comparison with the use of bevacizumab would be interesting.*

We appreciate the comments of the Reviewer and the suggestion that we compare the combination of SHP099 and AMG386 with VEGF blockade alone or in combination. It is indeed a topic of great interest to us and a focus of our next experiments.

24th Feb 2021

Dear Dr. Tosato,

Thank you for the submission of your manuscript to EMBO Molecular Medicine. We have now received feedback from reviewers #1 and #2, who had already reviewed the previous version of your manuscript. These reviewers also looked at your responses to referee #3's comments. As you will see from the reports below, while referee #2 is satisfied with the revisions and supports publication of the manuscript at this point, referee #1 would like to see the effects of anti-SHP2 on invasion and metastasis (already asked during the first round of review). Addressing this point will be necessary for further considering the manuscript in our journal, and acceptance of the manuscript will entail a second round of review.

EMBO Molecular Medicine encourages a single round of revision only and therefore, acceptance or rejection of the manuscript will depend on the completeness of your responses included in the next, final version of the manuscript. For this reason, and to save you from any frustrations in the end, I would strongly advise against returning an incomplete revision.

When submitting your revised manuscript, please carefully review the instructions that follow below. Failure to include requested items will delay the evaluation of your revision:

2) Individual production quality figure files as .eps, .tif, .jpg (one file per figure).

3) A .docx formatted letter INCLUDING the reviewers' reports and your detailed point-by-point responses to their comments. As part of the EMBO Press transparent editorial process, the point-by-point response is part of the Review Process File (RPF), which will be published alongside your paper.

4) A complete author checklist, which you can download from our author guidelines (<https://www.embopress.org/page/journal/17574684/authorguide#submissionofrevisions>). Please insert information in the checklist that is also reflected in the manuscript. The completed author checklist will also be part of the RPF.

6) Before submitting your revision, primary datasets produced in this study need to be deposited in an appropriate public database (see <https://www.embopress.org/page/journal/17574684/authorguide#dataavailability>).

The accession numbers and database should be listed in a formal "Data Availability" section (placed after Materials & Method). Please note that the Data Availability Section is restricted to

new primary data that are part of this study.

7) We would also encourage you to include the source data for figure panels that show essential data. Numerical data should be provided as individual .xls or .csv files (including a tab describing the data). For blots or microscopy, uncropped images should be submitted (using a zip archive if multiple images need to be supplied for one panel). Additional information on source data and instruction on how to label the files are available at .

8) Our journal encourages inclusion of *data citations in the reference list* to directly cite datasets that were re-used and obtained from public databases. Data citations in the article text are distinct from normal bibliographical citations and should directly link to the database records from which the data can be accessed. In the main text, data citations are formatted as follows: "Data ref: Smith et al, 2001" or "Data ref: NCBI Sequence Read Archive PRJNA342805, 2017". In the Reference list, data citations must be labeled with "[DATASET]". A data reference must provide the database name, accession number/identifiers and a resolvable link to the landing page from which the data can be accessed at the end of the reference. Further instructions are available at .

9) We replaced Supplementary Information with Expanded View (EV) Figures and Tables that are collapsible/expandable online. A maximum of 5 EV Figures can be typeset. EV Figures should be cited as 'Figure EV1, Figure EV2' etc... in the text and their respective legends should be included in the main text after the legends of regular figures.

- Additional Tables/Datasets should be labeled and referred to as Table EV1, Dataset EV1, etc. Legends have to be provided in a separate tab in case of .xls files. Alternatively, the legend can be supplied as a separate text file (README) and zipped together with the Table/Dataset file. See detailed instructions here:

10) The paper explained: EMBO Molecular Medicine articles are accompanied by a summary of the articles to emphasize the major findings in the paper and their medical implications for the non-specialist reader. Please provide a draft summary of your article highlighting

11) For more information: There is space at the end of each article to list relevant web links for further consultation by our readers. Could you identify some relevant ones and provide such information as well? Some examples are patient associations, relevant databases, OMIM/proteins/genes links, author's websites, etc...

12) Every published paper now includes a 'Synopsis' to further enhance discoverability. Synopses are displayed on the journal webpage and are freely accessible to all readers. They include a short stand first (maximum of 300 characters, including space) as well as 2-5 one-sentences bullet points that summarizes the paper. Please write the bullet points to summarize the key NEW findings. They should be designed to be complementary to the abstract - i.e. not repeat the same text. We encourage inclusion of key acronyms and quantitative information (maximum of 30 words / bullet point). Please use the passive voice. Please attach these in a separate file or send them by email, we will incorporate them accordingly.

Please also suggest a striking image or visual abstract to illustrate your article as a png file 550 px-wide x 400-px high.

13) As part of the EMBO Publications transparent editorial process initiative (see our Editorial at <http://embomolmed.embopress.org/content/2/9/329>), EMBO Molecular Medicine will publish online a Review Process File (RPF) to accompany accepted manuscripts.

In the event of acceptance, this file will be published in conjunction with your paper and will include the anonymous referee reports, your point-by-point response and all pertinent correspondence relating to the manuscript. Let us know whether you agree with the publication of the RPF and as here, if you want to remove or not any figures from it prior to publication.

I look forward to receiving your revised manuscript.

Yours sincerely,

Lise Roth

Lise Roth, PhD
Editor
EMBO Molecular Medicine

To submit your manuscript, please follow this link:

Link Not Available

Photos 400-800 DPI

*Additional important information regarding figures and illustrations can be found at <https://bit.ly/EMBOPressFigurePreparationGuideline>

***** Reviewer's comments *****

Referee #1 (Comments on Novelty/Model System for Author):

Due to the observed reduction of tumor vascularity and blood perfusion, the impact of the proposed anti-SHP2 therapy on cancer invasion and metastatic dissemination must be evaluated.

Referee #1 (Remarks for Author):

The revised version of the manuscript by Wang et al. is significantly improved, yet key issues still need to be addressed. Besides the vascular normalizing approach for cancer therapy, hypoxia is a key driver of tumor metastases (Golkes, Semenza, and Wirtz, 2014, Nat. Rev. Cancer 14:430-439; Schito & Semenza, 2016, Trends Cancer, 2:758-770). Hence, novel therapeutic approaches that reduce tumor vascularity and blood perfusion, such as the SHP2 phosphatase blockade proposed by Wang and collaborators, need to be checked at least for their ability to promote or not invasion and metastatic dissemination. As I suggested before, for example BF16F10 cell injection in the footpad may be used.

Referee #2 (Remarks for Author):

Criticisms are well addressed

Referee #1 (Comments on Novelty/Model System for Author):

Due to the observed reduction of tumor vascularity and blood perfusion, the impact of the proposed anti-SHP2 therapy on cancer invasion and metastatic dissemination must be evaluated.

Referee #1 (Remarks for Author):

The revised version of the manuscript by Wang et al. is significantly improved, yet key issues still need to be addressed. Besides the vascular normalizing approach for cancer therapy, hypoxia is a key driver of tumor metastases (Golkes, Semenza, and Wirtz, 2014, Nat. Rev. Cancer 14:430-439; Schito & Semenza, 2016, Trends Cancer, 2:758-770). Hence, novel therapeutic approaches that reduce tumor vascularity and blood perfusion, such as the SHP2 phosphatase blockade proposed by Wang and collaborators, need to be checked at least for their ability to promote or not invasion and metastatic dissemination. As I suggested before, for example B16F10 cell injection in the footpad may be used.

Response to Referee #1

We thank the Reviewer for the time and effort in the re-review of our manuscript, and for finding that the revised manuscript is “significantly improved”. We are appreciative.

We have now addressed the important question raised by the Reviewer, i.e. whether SHP2 inhibition drives increased metastatic dissemination. As suggested by the Reviewer, we have applied the model of B16F10 melanoma cell injection in the footpad of syngeneic mice and examined tumor cell dissemination to the ipsilateral popliteal lymph nodes and to the lungs. We have engaged the help of an expert in this model, Dr. Yanlin Yu, now a co-author.

The new results are presented in Figure 5, panels I-N, EV-4, panel C and related legends. The results are described in the text of the Results section (highlighted sections on pages 18 and 19). In brief, the results show that SHP2 inhibition did not increase metastatic dissemination of B16F10 cells. In the control group, 100% of mice displayed microscopic metastases in the ipsilateral popliteal lymph nodes; in the SHP099-treated group, 70% of mice displayed microscopic metastasis.

21st Apr 2021

Dear Dr. Tosato,

Thank you for the submission of your revised manuscript to EMBO Molecular Medicine. We have now received the enclosed report from the referee who re-reviewed your manuscript. As you will see, this referee is supportive of publication, and I am therefore pleased to inform you that we will be able to accept your manuscript, once the following editorial points will be addressed:

1) Main manuscript text:

- Please answer/correct the changes suggested by our data editors in the main manuscript file (in track changes mode). This file will be sent to you in the next couple of days. Please use this file for any further modification.
- Please remove the highlights in the text.
- Please remove "Data not shown" on pages 9, 11, 12, 17, 23: As per our guidelines, all data referred to in the paper should be displayed in the main or Expanded View figures.
- Please move the Material and Methods section after the Discussion. Please include in the main manuscript file most of the methods currently described in the Appendix (such as "flow cytometry", "animal experiments", "ELISAs and cell death proteomic assay", "gene expression"). Regarding human samples, please also include a statement that informed consent was obtained from all subjects and that the experiments conformed to the principles set out in the WMA Declaration of Helsinki and the Department of Health and Human Services Belmont Report.
- Please add a "Data availability section": the primary datasets produced in this study need to be deposited in an appropriate public database and the accession numbers added to this section. If no new dataset was produced, please indicate: "This study includes no data deposited in external repositories"
- Thank you for providing a Conflict of Interest section. Please move it after Author Contributions.
- Please make sure that the funding listed in the manuscript and in the submission system match (in our submission system, the funding is listed as HHS | National Institutes of Health (NIH) grant ZIA SC o10355, but in the manuscript, as the intramural program of the Center for Cancer Research, NC).
- Please update the reference format. References should be listed in alphabetical order and 10 authors should be listed before et al.

2) Figures, tables and appendix:

- Statistics: Please indicate in the figures or in the legends the exact $n=$ and exact $p=$ values along with the statistical test used. You may provide these values as a supplemental table in the Appendix file.
- Fig EV3 is mislabeled as Fig EV4#, please correct.
- Appendix: please define the arrowheads in Fig. S1

3) Checklist:

- Section E/12: please include a statement that informed consent was obtained from all subjects and that the experiments conformed to the principles set out in the WMA Declaration of Helsinki and the Department of Health and Human Services Belmont Report.
- Section E/14: if there are no restrictions on the availability of human samples (as it seems to be the case), please indicate N/A.

4) Thank you for providing Source Data. Please upload them as 1 PDF file per figure.

5) Thank you for providing The Paper Explained. I added minor edits to make it shorter, please let me know if you agree with the following:

Problem

Tumor tissues are dependent upon an adequate vascularization to support progressive expansion and local invasion. Despite considerable efforts at targeting the tumor vasculature to limit the blood supply to cancer tissues, therapeutic benefits have been limited. Increased understanding of endothelial and vascular biology provides new impetus for the anti-angiogenic treatment of cancer.

Results

Here, we have identified the enzyme SHP2 as an essential factor that supports endothelial cell survival and growth in the remodeling tumor vasculature, but not in the normal resting vasculature. Since endothelial cells form the inner layer of all vessels, extensive damage to endothelial cells impairs vessel function. In melanoma and carcinoma mouse tumor models, we show that systemic treatment with an inhibitor of SHP2, SHP099, promotes degeneration of the tumor vasculature, reduces the number of tumor vessels and tumor oxygenation, and impairs tumor growth. By combining SHP099 with AMG386, which blocks the Angiotensin/AKT cascade, the vascular and anti-tumor effects of SHP099 are amplified. Since SHP2 and the Angiotensin/AKT pathways are active in the vascular endothelial cells of human melanoma and colon carcinoma, SHP2 inhibitors alone or with Angiotensin/AKT inhibitors hold promise as new anti-vascular cancer drugs.

Impact

The current results show that inhibition of SHP2 represents a new and effective strategy to target endothelial cells in the tumor vasculature of mice without altering the resting normal vasculature. SHP2 inhibition could be particularly effective when applied more broadly to tumors where SHP2 is pro-oncogenic.

6) As part of the EMBO Publications transparent editorial process initiative (see our Editorial at <http://embomolmed.embopress.org/content/2/9/329>), EMBO Molecular Medicine will publish online a Review Process File (RPF) to accompany accepted manuscripts.

This file will be published in conjunction with your paper and will include the anonymous referee reports, your point-by-point response and all pertinent correspondence relating to the manuscript. Let us know whether you agree with the publication of the RPF.

I look forward to receiving your revised manuscript.

Yours sincerely,

Lise Roth

Lise Roth, PhD
Editor
EMBO Molecular Medicine

To submit your manuscript , please follow this link:

Link Not Available

The system will prompt you to fill in your funding and payment information. This will allow Wiley to send you a quote for the article processing charge (APC) in case of acceptance. This quote takes into account any reduction or fee waivers that you may be eligible for. Authors do not need to pay any fees before their manuscript is accepted and transferred to our publisher.

***** Reviewer's comments *****

Referee #1 (Remarks for Author):

The Authors have satisfactorily addressed all my concerns.

The authors performed the requested editorial changes.

4th May 2021

We are pleased to inform you that your manuscript is accepted for publication and is now being sent to our publisher to be included in the next available issue of EMBO Molecular Medicine.

Corresponding Author Name: Giovanna Tosato

Manuscript Number: EMM-2021-14089